# Algorithmic Analysis and Statistical Estimation of SLOPE via Approximate Message Passing

Zhiqi Bu[*]    Jason M. Klusowski[†]    Cynthia Rush[‡]    Weijie Su[§]

## Abstract

SLOPE is a relatively new convex optimization procedure for high-dimensional linear regression via the sorted $\ell_1$ penalty: the larger the rank of the fitted coefficient, the larger the penalty. This non-separable penalty renders many existing techniques invalid or inconclusive in analyzing the SLOPE solution. In this paper, we develop an asymptotically exact characterization of the SLOPE solution under Gaussian random designs through solving the SLOPE problem using approximate message passing (AMP). This algorithmic approach allows us to approximate the SLOPE solution via the much more amenable AMP iterates. Explicitly, we characterize the asymptotic dynamics of the AMP iterates relying on a recently developed state evolution analysis for non-separable penalties, thereby overcoming the difficulty caused by the sorted $\ell_1$ penalty. Moreover, we prove that the AMP iterates converge to the SLOPE solution in an asymptotic sense, and numerical simulations show that the convergence is surprisingly fast. Our proof rests on a novel technique that specifically leverages the SLOPE problem. In contrast to prior literature, our work not only yields an asymptotically sharp analysis but also offers an algorithmic, flexible, and constructive approach to understanding the SLOPE problem.

## 1  Introduction

Consider observing linear measurements $\boldsymbol{y} \in \mathbb{R}^n$ that are modeled by the equation

$$\boldsymbol{y} = \boldsymbol{X}\boldsymbol{\beta} + \boldsymbol{w}, \tag{1.1}$$

where $\boldsymbol{X} \in \mathbb{R}^{n \times p}$ is a known measurement matrix, $\boldsymbol{\beta} \in \mathbb{R}^p$ is an unknown signal, and $\boldsymbol{w} \in \mathbb{R}^n$ is the measurement noise. Among numerous methods that seek to recover the signal $\boldsymbol{\beta}$ from the observed data, especially in the setting where $\boldsymbol{\beta}$ is sparse and $p$ is larger than $n$, SLOPE has recently emerged as a useful procedure that allows for estimation and model selection [9]. This method reconstructs the signal by solving the minimization problem

$$\widehat{\boldsymbol{\beta}} := \arg\min_{\boldsymbol{b}} \frac{1}{2}\|\boldsymbol{y} - \boldsymbol{X}\boldsymbol{b}\|^2 + \sum_{i=1}^{p} \lambda_i |\boldsymbol{b}|_{(i)}, \tag{1.2}$$

where $\|\cdot\|$ denotes the $\ell_2$ norm, $\lambda_1 \geq \cdots \geq \lambda_p \geq 0$ (with at least one strict inequality) is a sequence of thresholds, and $|\boldsymbol{b}|_{(1)} \geq \cdots \geq |\boldsymbol{b}|_{(p)}$ are the order statistics of the fitted coefficients

---

[*]Department of Applied Mathematics and Computational Science, University of Pennsylvania, Philadelphia, PA 19104, USA. Email: zbu@sas.upenn.edu

[†]Department of Statistics, Rutgers University, New Brunswick, NJ 08854, USA. Email: jason.klusowski@rutgers.edu  Supported in part by NSF DMS #1915932.

[‡]Department of Statistics, Columbia University, New York, NY 10027, USA. Email: cynthia.rush@columbia.edu  Supported in part by NSF CCF #1849883.

[§]Department of Statistics, University of Pennsylvania, Philadelphia, PA 19104, USA. Email: suw@wharton.upenn.edu Supported in part by NSF DMS CAREER #1847415 and NSF CCF #1763314.

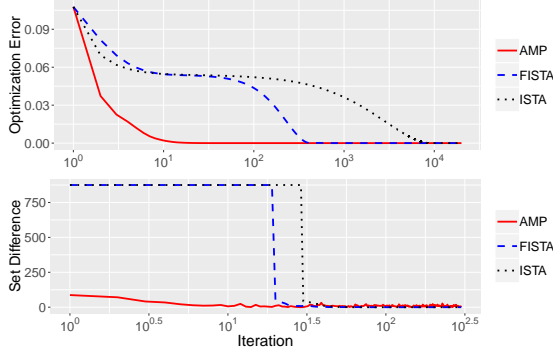

**Figure 1:** Optimization errors, $||\boldsymbol{\beta}^t - \widehat{\boldsymbol{\beta}}||^2/p$, and (symmetric) set difference of $\mathrm{supp}(\boldsymbol{\beta}^t)$ and $\mathrm{supp}(\widehat{\boldsymbol{\beta}})$.

| | | Optimization errors | | | | |
|---|---|---|---|---|---|---|
| | Set Diff | $10^{-2}$ | $10^{-3}$ | $10^{-4}$ | $10^{-5}$ | $10^{-6}$ |
| ISTA | 60 | 4048 | 7326 | 8569 | 9007 | 9161 |
| FISTA | 47 | 275 | 374 | 412 | 593 | 604 |
| AMP | 30 | 6 | 13 | 22 | 32 | 40 |

**Table 1:** First iteration $t$ for which there is zero set difference or optimization error $||\boldsymbol{\beta}^t - \widehat{\boldsymbol{\beta}}||^2/p$ falls below a threshold.

Setting of Figure 1 and Table 1: Design $X$ is $500 \times 1000$ and has i.i.d. $\mathcal{N}(0, 1/500)$ entries. True signal $\boldsymbol{\beta}$ is elementwise i.i.d. Gaussian-Bernoulli: $\mathcal{N}(0, 1)$ with probability 0.1 and 0 otherwise. Noise variance $\sigma_w^2 = 0$. A careful calibration between the thresholds $\boldsymbol{\theta}_t$ in AMP and $\boldsymbol{\lambda}$ is SLOPE is used. Details in Section 2.

in absolute value. The regularizer $\sum \lambda_i |\boldsymbol{b}|_{(i)}$ is a *sorted $\ell_1$-norm* (denoted as $J_{\boldsymbol{\lambda}}(\boldsymbol{b})$ henceforth), which is *non-separable* due to the sorting operation involved in its calculation. Notably, SLOPE has two attractive features that are not simultaneously present in other methods for linear regression including the LASSO [34] and knockoffs [1]. Explicitly, on the estimation side, SLOPE achieves minimax estimation properties under certain random designs *without* requiring any knowledge of the sparsity degree of $\boldsymbol{\beta}$ [32, 7]. On the testing side, SLOPE controls the false discovery rate in the case of independent predictors [9, 12]. For completeness, we remark that [10, 35, 20] proposed similar non-separable regularizers to encourage grouping of correlated predictors.

This work is concerned with the algorithmic aspects of SLOPE through the lens of *approximate message passing* (AMP) [2, 17, 22, 24, 27]. AMP is a class of computationally efficient and easy-to-implement algorithms for a broad range of statistical estimation problems, including compressed sensing and the LASSO [3]. When applied to SLOPE, AMP takes the following form: at initial iteration $t = 0$, assign $\boldsymbol{\beta}^0 = \mathbf{0}$, $\boldsymbol{z}^0 = \boldsymbol{y}$, and for $t \geq 0$,

$$\boldsymbol{\beta}^{t+1} = \mathrm{prox}_{J_{\boldsymbol{\theta}_t}}(\boldsymbol{X}^\top \boldsymbol{z}^t + \boldsymbol{\beta}^t), \tag{1.3a}$$

$$\boldsymbol{z}^{t+1} = \boldsymbol{y} - \boldsymbol{X}\boldsymbol{\beta}^{t+1} + \frac{\boldsymbol{z}^t}{n}\left[\nabla \mathrm{prox}_{J_{\boldsymbol{\theta}_t}}(\boldsymbol{X}^\top \boldsymbol{z}^t + \boldsymbol{\beta}^t)\right]. \tag{1.3b}$$

The non-increasing sequence $\boldsymbol{\theta}_t$ is proportional to $\boldsymbol{\lambda}$ and will be given explicitly in Section 2. Here, $\mathrm{prox}_{J_{\boldsymbol{\theta}}}$ is the proximal operator of the sorted $\ell_1$ norm, that is,

$$\mathrm{prox}_{J_{\boldsymbol{\theta}}}(\boldsymbol{x}) := \operatorname*{argmin}_{\boldsymbol{b}} \frac{1}{2}\|\boldsymbol{x} - \boldsymbol{b}\|^2 + J_{\boldsymbol{\theta}}(\boldsymbol{b}),$$

and $\nabla \mathrm{prox}_{J_{\boldsymbol{\theta}}}$ denotes the divergence of the proximal operator (see an equivalent but more explicit form of this algorithm in Section 2 and other preliminaries on SLOPE and the prox operator defined above in Appendix A). Compared to the proximal gradient descent (ISTA) [15, 16, 26], AMP has an extra correction term in its residual step that adjusts the iteration in a non-trivial way and seeks to provide improved convergence performance [17, 11].

The *empirical* performance of AMP in solving SLOPE is illustrated in Figure 1 and Table 1, which suggest the superiority of AMP over ISTA and FISTA [6]—perhaps the two most popular proximal gradient descent methods—in terms of speed of convergence. However, the vast AMP literature thus far remains silent on whether AMP *provably* solves SLOPE and, if so, whether one can leverage AMP to get insights into the statistical properties of SLOPE. This vacuum in the literature is due to the *non-separability* of the SLOPE regularizer, making it a major challenge to apply AMP to SLOPE directly. In stark contrast, AMP theory has been rigorously applied to the LASSO [3], showing both good empirical performance and nice theoretical properties of solving the LASSO using AMP. Moreover, AMP in this setting allows for asymptotically exact statistical characterization of its output, which converges to the LASSO solution, thereby providing a powerful tool in fine-grained analyses of the LASSO [4, 33, 25].

In this work, we prove that the AMP algorithm (1.3) solves the SLOPE problem in an asymptotically *exact* sense under independent Gaussian random designs. Our proof uses the recently extended

AMP theory for non-separable denoisers [8] and applies this tool to derive the state evolution that describes the asymptotically exact behaviors of the AMP iterates $\boldsymbol{\beta}^t$ in (1.3). The next step, which is the core of our proof, is to relate the AMP estimates to the SLOPE solution. This presents several challenges that *cannot* be resolved only within the AMP framework. In particular, unlike the LASSO, the number of nonzeros in the SLOPE solution can exceed the number of observations. This fact imposes substantially more difficulties on showing that the distance between the SLOPE solution and the AMP iterates goes to zero than in the LASSO case due to the possible *non-strong convexity* of the SLOPE problem, even restricted to the solution support. To overcome these challenges, we develop novel techniques that are tailored to the characteristics of the SLOPE solution. For example, our proof relies on the crucial property of SLOPE that the *unique* nonzero components of its solution never outnumber the observation units.

As a byproduct, our analysis gives rise to an *exact* asymptotic characterization of the SLOPE solution under independent Gaussian random designs through leveraging the statistical aspect of the AMP theory. In slightly more detail, the probability distribution of the SLOPE solution is completely specified by a few parameters that are the solution to a certain fixed-point equation in an asymptotic sense. This provides a powerful tool for fine-grained statistical analysis of SLOPE as it was for the LASSO problem. We note that a recent paper [21]—which takes an entirely different path—gives an asymptotic characterization of the SLOPE solution that matches our asymptotic analysis that is deduced from our AMP theory for SLOPE. However, our AMP-based approach is more algorithmic in nature and offers a more concrete connection between the finite-sample behaviors of the SLOPE problem and its asymptotic distribution via the computationally efficient AMP algorithm.

## 2 Algorithmic Development

In this section we develop an AMP algorithm for finding the SLOPE estimator in (1.2). Recall the AMP algorithm we study is (1.3). Specifically, it is through the threshold values $\boldsymbol{\theta}_t$ that one can ensure the AMP estimates converge to the SLOPE estimator with parameter $\boldsymbol{\lambda}$. In this section we present how one should calibrate the thresholds of the AMP iterations in (1.3) in order for the algorithm to solve SLOPE cost in (1.2). Then in Section 3, we prove rigorously that the AMP algorithm solves the SLOPE optimization asymptotically and we leverage theoretical guarantees for the AMP algorithm to exactly characterize the mean square error of the SLOPE estimator in the large system limit. This is done by applying recent theoretical results for AMP algorithms that use a non-separable non-linearity [8], like the one in (1.3).

We first note that the analysis we pursue in this work makes the following assumptions about the linear model (1.1) and parameter vector in (A.1):

- **(A1)** The measurement matrix $\boldsymbol{X}$ has independent and identically-distributed (i.i.d.) Gaussian entries that have mean $0$ and variance $1/n$.

- **(A2)** The signal $\boldsymbol{\beta}$ has elements that are i.i.d. $B$, with $\mathbb{E}(B^2 \max\{0, \log B\}) < \infty$.

- **(A3)** The noise $\boldsymbol{w}$ is elementwise i.i.d. $W$, with $\sigma_w^2 := \mathbb{E}(W^2) < \infty$.

- **(A4)** The vector $\boldsymbol{\lambda}(p) = (\lambda_1, \dots, \lambda_p)$ is elementwise i.i.d. $\Lambda$, with $\mathbb{E}(\Lambda^2) < \infty$.

- **(A5)** The ratio $n/p$ approaches a constant $\delta \in (0, \infty)$ in the large system limit, as $n, p \to \infty$.

**Remark: (A4)** can be relaxed as $\lambda_1, \dots, \lambda_p$ having an empirical distribution that converges weakly to probability measure $\Lambda$ on $\mathbb{R}$ with $\mathbb{E}(\Lambda^2) < \infty$ and $\|\boldsymbol{\lambda}(p)\|^2/p \to \mathbb{E}(\Lambda^2)$. A similar relaxation can be made for assumptions **(A2)** and **(A3)**.

### 2.1 SLOPE Preliminaries

For a vector $\boldsymbol{v} \in \mathbb{R}^p$, the divergence of the proximal operator, $\nabla \operatorname{prox}_f(\boldsymbol{v})$, is given by the following:

$$\nabla \operatorname{prox}_f(\boldsymbol{v}) := \sum_{i=1}^p \frac{\partial}{\partial v_i}[\operatorname{prox}_f(\boldsymbol{v})]_i = \left(\frac{\partial}{\partial v_1}, \frac{\partial}{\partial v_2}, \dots, \frac{\partial}{\partial v_p}\right) \cdot \operatorname{prox}_f(\boldsymbol{v}), \qquad (2.1)$$

where as given in [32], proof of Fact 3.4,

$$\frac{\partial [\mathrm{prox}_{J_{\boldsymbol{\lambda}}}(\boldsymbol{v})]_i}{\partial v_j} = \begin{cases} \frac{\mathrm{sign}([\mathrm{prox}_{J_{\boldsymbol{\lambda}}}(\boldsymbol{v})]_i) \cdot \mathrm{sign}([\mathrm{prox}_{J_{\boldsymbol{\lambda}}}(\boldsymbol{v})]_j)}{\#\{1 \leq k \leq p : |[\mathrm{prox}_{J_{\boldsymbol{\lambda}}}(\boldsymbol{v})]_k| = |[\mathrm{prox}_{J_{\boldsymbol{\lambda}}}(\boldsymbol{v})]_j|\}}, & \text{if } |[\mathrm{prox}_{J_{\boldsymbol{\lambda}}}(\boldsymbol{v})]_j| = |[\mathrm{prox}_{J_{\boldsymbol{\lambda}}}(\boldsymbol{v})]_i|, \\ 0, & \text{otherwise.} \end{cases} \tag{2.2}$$

Hence the divergence is simplified to

$$\nabla \mathrm{prox}_{J_{\boldsymbol{\lambda}}}(\boldsymbol{v}) = \| \mathrm{prox}_{J_{\boldsymbol{\lambda}}}(\boldsymbol{v})\|_0^*, \tag{2.3}$$

where $\| \cdot \|_0^*$ counts the unique non-zero magnitudes in a vector, e.g. $\|(0, 1, -2, 0, 2)\|_0^* = 2$. This explicit form of divergence not only waives the need to use approximation in calculation but also speed up the recursion, since it only depends on the proximal operator as a whole instead of on $\boldsymbol{\theta}_{t-1}, \boldsymbol{X}, \boldsymbol{z}^{t-1}, \boldsymbol{\beta}^{t-1}$. Therefore, we have

**Lemma 2.1.** *In AMP,* (1.3b) *is equivalent to*

$$\boldsymbol{z}^{t+1} = \boldsymbol{y} - \boldsymbol{X}\boldsymbol{\beta}^{t+1} + \frac{\boldsymbol{z}^t}{\delta p}\|\boldsymbol{\beta}^{t+1}\|_0^*.$$

Other preliminary ideas and background on SLOPE and the prox operator are found in Appendix A.

## 2.2 AMP Background

An attractive feature of AMP is that its statistical properties can be exactly characterized at each iteration $t$, at least asymptotically, via a one-dimensional recursion known as state evolution [2, 8]. Specifically, it can be shown that the pseudo-data, meaning the input $\boldsymbol{X}^{\top}\boldsymbol{z}^t + \boldsymbol{\beta}^t$ for the estimate of the unknown signal in (1.3a), is asymptotically equal in distribution to the true signal plus independent, Gaussian noise, i.e. $\boldsymbol{\beta} + \tau_t \boldsymbol{Z}$, where the noise variance $\tau_t$ is defined by the state evolution. For this reason, the function used to update the estimate in (1.3a), in our case, the proximal operator, $\mathrm{prox}_{J_{\boldsymbol{\theta}_t}}(\cdot)$, is usually referred to as a 'denoiser' in the AMP literature.

This statistical characterization of the pseudo-data was first rigorously shown to be true in the case of 'separable' denoisers by Bayati and Montanari [2], and an analysis of the rate of this convergence was given in [31]. A 'separable' denoiser is one that applies the same (possibly non-linear) function to each element of its input. Recent work, which we make use of in this paper, proves that asymptotically the pseudo-data has distribution $\boldsymbol{\beta} + \tau_t \boldsymbol{Z}$ when non-separable 'denoisers' are used in the AMP algorithm.

The dynamics of the AMP iterations are tracked by a recursive sequence referred to as the state evolution, defined below. For $\boldsymbol{B}$ elementwise i.i.d. $B$ independent of $\boldsymbol{Z} \sim \mathcal{N}(0, \mathbb{I}_p)$, let $\tau_0^2 = \sigma_w^2 + \mathbb{E}[B^2]/\delta$ and for $t \geq 0$,

$$\tau_{t+1}^2 = \sigma_w^2 + \lim_p \frac{1}{\delta p} \mathbb{E}\|\mathrm{prox}_{J_{\boldsymbol{\theta}_t}}(\boldsymbol{B} + \tau_t \boldsymbol{Z}) - \boldsymbol{B}\|^2. \tag{2.4}$$

Below we make rigorous the way that the recursion in (2.4) relates to the AMP iteration (1.3a)-(1.3b).

We note that throughout, we let $\mathcal{N}(\mu, \sigma^2)$ denote the Gaussian density with mean $\mu$ and variance $\sigma^2$ and we use $\mathbb{I}_p$ to indicate a $p \times p$ identity matrix.

## 2.3 Analysis of the AMP State Evolution

As mentioned previously, it is through the sequence of thresholds $\boldsymbol{\theta}_t$ that one is able to relate the AMP algorithm to the SLOPE estimator in (1.2) for certain $\boldsymbol{\lambda}$. Specifically, we will choose $\boldsymbol{\theta}_t = \boldsymbol{\alpha}\tau_t(p)$ for every iteration $t$ where the vector $\boldsymbol{\alpha}$ is fixed via a calibration made explicit below and $\tau_t^2(p)$ is defined using an approximation to the state evolution in (2.4) given in (2.5) below. We can interpret this to mean that within the AMP algorithm, $\boldsymbol{\alpha}$ plays the role of the regularizer $\boldsymbol{\lambda}$.

The calibration is motivated by a careful analysis of the following approximation (when $p$ is large) to the state evolution iteration in (2.4). Namely,

$$\tau_{t+1}^2(p) = \sigma_w^2 + \frac{1}{\delta p} \mathbb{E}\|\mathrm{prox}_{J_{\boldsymbol{\alpha}\tau_t(p)}}(\boldsymbol{\beta} + \tau_t(p)\boldsymbol{Z}) - \boldsymbol{\beta}\|^2, \tag{2.5}$$

where the difference between (2.5) and the state evolution (2.4) is via the large system limit in $p$. When we refer to the recursion in (2.5) we will always specify the $p$ dependence explicitly as $\tau_t(p)$.

Before we introduce this calibration, however, we give the following result which motivates why the AMP iteration should relate at all to the SLOPE estimator.

**Lemma 2.2.** *Any stationary point $\widehat{\beta}$ (with corresponding $\widehat{z}$) in the AMP algorithm* (1.3a)-(1.3b) *with $\theta_* = \alpha\tau_*$ is a minimizer of the SLOPE cost function in* (1.2) *with*

$$\boldsymbol{\lambda} = \boldsymbol{\theta}_*\Big(1 - \frac{1}{\delta p}\Big(\nabla\,\mathrm{prox}_{J_{\boldsymbol{\theta}_*}}(\widehat{\beta} + \boldsymbol{X}^\top\widehat{z})\Big)\Big) = \boldsymbol{\theta}_*\Big(1 - \frac{1}{n}\Big\|\mathrm{prox}_{J_{\boldsymbol{\theta}_*}}(\widehat{\beta} + \boldsymbol{X}^\top\widehat{z})\Big\|_0^*\Big).$$

*Proof of Lemma 2.2.* By stationarity,

$$\widehat{\beta} = \mathrm{prox}_{\boldsymbol{\theta}_*}(\widehat{\beta} + \boldsymbol{X}^\top\widehat{z}) \quad\text{and}\quad \widehat{z} = \boldsymbol{y} - \boldsymbol{X}\widehat{\beta} + \frac{\widehat{z}}{\delta p}(\nabla\,\mathrm{prox}_{\boldsymbol{\theta}_*}(\widehat{\beta} + \boldsymbol{X}^\top\widehat{z})). \tag{2.6}$$

Denote by $\omega := \frac{1}{\delta p}(\nabla\,\mathrm{prox}_{\boldsymbol{\theta}_*}(\widehat{\beta} + \boldsymbol{X}^\top\widehat{z}))$. Then, from (2.6), $\widehat{z} = \frac{\boldsymbol{y} - \boldsymbol{X}\widehat{\beta}}{1-\omega}$, and by (2.6) along with Fact A.1, $\boldsymbol{X}^\top\widehat{z} \in \partial J_{\boldsymbol{\theta}_*}(\widehat{\beta})$. Clearly, $\boldsymbol{X}^\top\widehat{z} = \frac{\boldsymbol{X}^\top(\boldsymbol{y} - \boldsymbol{X}\widehat{\beta})}{1-\omega} \in J_{\boldsymbol{\theta}_*}(\widehat{\beta})$, which tells us $\boldsymbol{X}^\top(\boldsymbol{y} - \boldsymbol{X}\widehat{\beta}) \in J_{\boldsymbol{\theta}_*(1-\omega)}(\widehat{\beta})$ which is *exactly* the stationary condition of SLOPE with $\boldsymbol{\lambda} = (1-\omega)\boldsymbol{\theta}_*$ as desired. $\square$

Results about the recursion (2.5) are summarized in the following theorem and the theorem's proof is given in Appendix C. We first introduce some useful notations: let $\boldsymbol{A}_{\min}(\delta)$ be the set of solutions to

$$\delta = f(\boldsymbol{\alpha}), \text{ where } f(\boldsymbol{\alpha}) := \frac{1}{p}\sum_{i=1}^p \mathbb{E}\left\{\Big(1 - |[\mathrm{prox}_{J_{\boldsymbol{\alpha}}}(\boldsymbol{Z})]_i|\sum_{j\in I_i}\alpha_j\Big)/[\boldsymbol{D}(\mathrm{prox}_{J_{\boldsymbol{\alpha}}}(\boldsymbol{Z}))]_i\right\} \tag{2.7}$$

Here $\odot$ represents elementwise multiplication of vectors and for a vector $\boldsymbol{v} \in \mathbb{R}^p$, $\boldsymbol{D}$ is defined elementwise as $[\boldsymbol{D}(\boldsymbol{v})]_i = \#\{j : |v_j| = |v_i|\}$ if $v_i \neq 0$ and $\infty$ otherwise. For $\boldsymbol{u} \in \mathbb{R}^m$, the notation $\langle\boldsymbol{u}\rangle := \sum_{i=1}^m u_i/m$ and we say a vector $\boldsymbol{u}$ is larger than $\boldsymbol{v}$ if $\forall i, u_i > v_i$. The expectation in (2.7) is taken with respect to $\boldsymbol{Z}$, a $p$-length vector of i.i.d. standard Gaussians.

**Theorem 1.** *For any $\boldsymbol{\alpha}$ strictly larger than at least one element in the set $\boldsymbol{A}_{\min}(\delta)$, the recursion in* (2.5) *has a unique fixed point and denoting this fixed point by $\tau_*^2(p)$. Then $\tau_t(p) \to \tau_*(p)$ for any initial condition and monotonically. Moreover, defining a function $F: \mathbb{R} \times \mathbb{R}^p \to \mathbb{R}$ as*

$$F(\tau^2(p), \boldsymbol{\alpha}\tau(p)) := \sigma^2 + \frac{1}{\delta p}\mathbb{E}\|\mathrm{prox}_{J_{\boldsymbol{\alpha}\tau(p)}}(\boldsymbol{B} + \tau(p)\boldsymbol{Z}) - \boldsymbol{B}\|^2, \tag{2.8}$$

*where $\boldsymbol{B}$ is elementwise i.i.d. $B$ independent of $\boldsymbol{Z} \sim \mathcal{N}(0, \mathbb{I}_p)$, so that $\tau_{t+1}^2(p) = F(\tau_t^2(p), \boldsymbol{\alpha}\tau_t(p))$, then $|\frac{\partial F}{\partial \tau^2(p)}(\tau^2(p), \boldsymbol{\alpha}\tau(p))| < 1$ at $\tau(p) = \tau_*(p)$. Moreover, for $f(\boldsymbol{\alpha})$ defined in* (2.7), *we show that $f(\boldsymbol{\alpha}) = \delta \lim_{\tau(p)\to\infty} dF/d\tau^2(p)$.*

Notice that Theorem 1 gives necessary conditions on the calibration vector $\boldsymbol{\alpha}$ under which recursion in (2.5), and equivalently, the calibration given below are well-defined.

## 2.4 Threshold Calibration

Motivated by Lemma 2.2 and Lemma B.1, we define a calibration from the regularization parameter $\boldsymbol{\lambda}$, to the corresponding threshold $\boldsymbol{\alpha}$ used to define the AMP algorithm. Such calibration is asymptotically exact when $p = \infty$.

In practice, we will be given finite-length $\boldsymbol{\lambda}$ and then we want to design the AMP iteration to solve the corresponding SLOPE cost. We do this by choosing $\boldsymbol{\alpha}$ as the vector that solves $\boldsymbol{\lambda} = \boldsymbol{\lambda}(\boldsymbol{\alpha})$ where

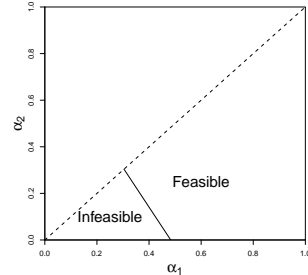

**Figure 2:** $\boldsymbol{A}_{\min}$ (black curve) when $p = 2$ and $\delta = 0.6$.

$$\boldsymbol{\lambda}(\boldsymbol{\alpha}) := \boldsymbol{\alpha}\tau_*(p)\Big(1 - \frac{1}{n}\mathbb{E}\|\,\mathrm{prox}_{J_{\boldsymbol{\alpha}\tau_*(p)}}(\boldsymbol{B} + \tau_*(p)\boldsymbol{Z})\|_0^*\Big), \tag{2.9}$$

where $\boldsymbol{B}$ is elementwise i.i.d. $B$ independent of $\boldsymbol{Z} \sim \mathcal{N}(0, \mathbb{I}_p)$ and $\tau_*(p)$ is the limiting value defined in Theorem 1. We note the fact that the calibration in (2.9) sets $\boldsymbol{\alpha}$ as a vector *in the same direction* as $\boldsymbol{\lambda}$, but that is scaled by a constant value (for each $p$), where the constant value is given by $\tau_*(p)(1 - \mathbb{E} \| \operatorname{prox}_{J_{\boldsymbol{\alpha}\tau_*(p)}}(\boldsymbol{B} + \tau_*(p)\boldsymbol{Z})\|_0^*)/n$.

We claim that the calibration (2.9) and its inverse $\boldsymbol{\lambda} \mapsto \boldsymbol{\alpha}(\boldsymbol{\lambda})$ are well-defined. In [3, Proposition 1.4 (first introduced in [18]) and Corollary 1.7] this is proved rigorously for the LASSO calibration and we claim that this proof can be adapted to the present case without many difficulties, though we don't pursue this in the current document.

**Proposition 2.3.** *The function $\boldsymbol{\alpha} \mapsto \boldsymbol{\lambda}(\boldsymbol{\alpha})$ defined in (2.9) is continuous on $\{\boldsymbol{\alpha} : f(\boldsymbol{\alpha}) < \delta\}$ for $f(\cdot)$ defined in (2.7) with $\boldsymbol{\lambda}(\boldsymbol{A}_{\min}) = -\infty$ and $\lim_{\boldsymbol{\alpha} \to \infty} \boldsymbol{\lambda}(\boldsymbol{\alpha}) = \infty$ (where the limit is taken elementwise). Therefore the inverse function $\boldsymbol{\lambda} \mapsto \boldsymbol{\alpha}(\boldsymbol{\lambda})$ exists and is continuous non-decreasing for any $\boldsymbol{\lambda} > \boldsymbol{0}$.*

This proposition motivates Algorithm 1 which uses bisection method to find the unique $\boldsymbol{\alpha}$ for each $\boldsymbol{\lambda}$. It suffices to find two guesses of $\boldsymbol{\alpha}$ parallel to $\boldsymbol{\lambda}$ that, when mapped via (2.9), sandwich the true $\boldsymbol{\lambda}$. The proof of this proposition can be found in [13, Appendix A.2].

---

**Algorithm 1** Calibration from $\boldsymbol{\lambda} \to \boldsymbol{\alpha}$

---

1. Initialize $\alpha_1 = \alpha_{\min}$ such that $\alpha_{\min}\boldsymbol{\ell} \in \boldsymbol{A}_{\min}$, where $\boldsymbol{\ell} := \boldsymbol{\lambda}/\lambda_1$; Initialize $\alpha_2 = 2\alpha_1$
**while** $L(\alpha_2) < 0$ where $L : \mathbb{R} \to \mathbb{R}; \alpha \mapsto \operatorname{sign}(\boldsymbol{\lambda}(\alpha\boldsymbol{\ell}) - \boldsymbol{\lambda})$ **do**
   2. Set $\alpha_1 = \alpha_2, \alpha_2 = 2\alpha_2$
**end while**
3. **return BISECTION** $(L(\alpha), \alpha_1, \alpha_2)$

---

Remark: $\operatorname{sign}(\boldsymbol{\lambda}(\cdot) - \boldsymbol{\lambda}) \in \mathbb{R}$ is well-defined since $\boldsymbol{\lambda}(\cdot) \parallel \boldsymbol{\lambda}$ implies all entries share the same sign. The function "**BISECTION**$(L, a, b)$" finds the root of $L$ in $[a, b]$ via the bisection method.

---

As noted previously, the calibration in (2.9) is exact when $p \to \infty$, so we study the mapping between $\boldsymbol{\alpha}$ and $\boldsymbol{\lambda}$ in this limit. Recall from **(A4)**, that the sequence of vectors $\{\boldsymbol{\lambda}(p)\}_{p \geq 0}$ are drawn i.i.d. from distribution $\Lambda$. It follows that the sequence $\{\boldsymbol{\alpha}(p)\}_{p \geq 0}$ defined for each $p$ by the finite-sample calibration (2.9) are i.i.d. from a distribution $A$, where $A$ satisfies $\mathbb{E}(A^2) < \infty$, and is defined via

$$\Lambda = A\tau_* \Big( 1 - \lim_p \frac{1}{\delta p} \mathbb{E} \| \operatorname{prox}_{J_{\boldsymbol{A}(p)\tau_*}}(\boldsymbol{B} + \tau_*\boldsymbol{Z})\|_0^* \Big), \tag{2.10}$$

where $\boldsymbol{A}(p) \in \mathbb{R}^p$ are order statistics of $p$ i.i.d. draws from $A$ given by (2.10) and $\tau_*$ is defined as the large $t$ limit of (2.4). We note that the calibrations presented in this section are well-defined:

**Fact 2.4.** *The limits in (2.4) and (2.10) exist.*

This fact is proven in Appendix E. One idea used in the proof is that the prox operator is *asymptotically separable*, a result shown by [21, Proposition 1]. Specifically, for sequences of input, $\{\boldsymbol{v}(p)\}$, and thresholds, $\{\boldsymbol{\lambda}(p)\}$, both having empirical distributions that weakly converge to a distributions $V$ and $\Lambda$, respectively, then there exists a limiting scalar function $h(\cdot) := h(\boldsymbol{v}(p); V, \Lambda)$ (determined by $V$ and $\Lambda$) of the proximal operator $\operatorname{prox}_{J_{\boldsymbol{\lambda}}}(\boldsymbol{v}(p))$. Further details are shown in Appendix E, Lemma E.1. Using $h(\cdot) := h(\cdot; B + \tau_*Z, A\tau_*)$, this argument implies that (2.4) can be represented as

$$\tau_*^2 := \sigma^2 + \mathbb{E}(h(B + \tau_*Z) - B)^2/\delta,$$

and if we denote $m$ as the Lebesgue measure, then the limit in (2.10) can be represented as

$$\mathbb{P}\Big( B + \tau_*Z \in \Big\{ x \mid h(x) \neq 0 \quad \text{and} \quad m\{z \mid |h(z)| = |h(x)|\} = 0 \Big\} \Big).$$

In other words, the limit in (2.10) is the Lebesgue measure of the domain of the quantile function of $h$ for which the quantile of $h$ assumes unique values (i.e., is not flat).

## 3 Asymptotic Characterization of SLOPE

### 3.1 AMP Recovers the SLOPE Estimate

Here we show that the AMP algorithm converges in $\ell_2$ to the SLOPE estimator, implying that the AMP iterates can be used as a surrogate for the global optimum of the SLOPE cost function. The

schema of the proof is similar to [3, Lemma 3.1], however, major differences lie in the fact that the proximal operator used in the AMP updates (1.3a)-(1.3b) is non-separable. We sketch the proof here, and a forthcoming article will be devoted to giving a complete and detailed argument.

**Theorem 2.** *Under assumptions (A1) - (A5), for the output of the AMP algorithm in* (1.3a) *and the SLOPE estimate* (1.2),

$$\operatorname*{plim}_{p \to \infty} \frac{1}{p} \|\widehat{\boldsymbol{\beta}} - \boldsymbol{\beta}^t\|^2 = c_t, \quad where \quad \lim_{t \to \infty} c_t = 0. \tag{3.1}$$

*Proof.* The proof requires dealing carefully with the fact that the SLOPE cost function given in (1.2) is *not* necessarily strongly convex, meaning that we could encounter the undesirable situation where $\mathcal{C}(\widehat{\boldsymbol{\beta}})$ is close to $\mathcal{C}(\boldsymbol{\beta})$ but $\widehat{\boldsymbol{\beta}}$ is not close to $\boldsymbol{\beta}$, meaning the statistical recovery of $\boldsymbol{\beta}$ would be poor.

In the LASSO case, one works around this challenge by showing that the (LASSO) cost function does have nice properties when considering just the elements of the non-zero support of $\boldsymbol{\beta}^t$ at any (large) iteration $t$. In the LASSO case, the non-zero support of $\boldsymbol{\beta}$ has size no larger than $n < p$.

In the SLOPE problem, however, it is possible that the support set has size exceeding $n$, and therefore the LASSO analysis is not immediately applicable. Our proof develops novel techniques that are tailored to the characteristics of the SLOPE solution. Specifically, when considering the SLOPE problem, one can show nice properties (similar to those in the LASSO case) by considering a support-like set, that being the *unique* non-zeros in the estimate $\boldsymbol{\beta}^t$ at any (large) iteration $t$. In other words, if we define an equivalence relation $x \sim y$ when $|x| = |y|$, then entries of AMP estimate at any iteration $t$ are partitioned into equivalence classes. Then we observe from (2.9), and the non-negativity of $\boldsymbol{\lambda}$, that the number of equivalence classes is no larger than $n$. We see an analogy between SLOPE's equivalence class (or 'maximal atom' as described in Appendix A) and LASSO's support set. This approach allows us to deal with the lack of a strongly convex cost. $\quad\square$

Theorem 2 ensures that the AMP algorithm solves the SLOPE problem in an asymptotic sense. To better appreciate the convergence guarantee, it calls for elaboration on (3.1). First, it implies that $\|\widehat{\boldsymbol{\beta}} - \boldsymbol{\beta}^t\|^2/p$ converges in probability to a constant, say $c_t$. Next, (3.1) says that $c_t \to 0$ as $t \to \infty$.

## 3.2 Exact Asymptotic Characterization of the SLOPE Estimate

A consequence of Theorem B.1, is that the SLOPE estimator $\widehat{\boldsymbol{\beta}}$ inherits performance guarantees provided by the AMP state evolution, in the sense of Theorem 3 below. Theorem 3 provides as asymptotic characterization of pseudo-Lipschitz loss between $\widehat{\boldsymbol{\beta}}$ and the truth $\boldsymbol{\beta}$.

**Definition 3.1. Uniformly pseudo-Lipschitz functions** *[8]: For $k \in \mathbb{N}_{>0}$, a function $\phi : \mathbb{R}^d \to \mathbb{R}$ is* pseudo-Lipschitz *of order $k$ if there exists a constant L, such that for $\boldsymbol{a}, \boldsymbol{b} \in \mathbb{R}^d$,*

$$\|\phi(\boldsymbol{a}) - \phi(\boldsymbol{b})\| \leq L\Big(1 + (\|\boldsymbol{a}\|/\sqrt{d})^{k-1} + (\|\boldsymbol{b}\|/\sqrt{d})^{k-1}\Big)\Big(\|\boldsymbol{a} - \boldsymbol{b}\|/\sqrt{d}\Big). \tag{3.2}$$

*A sequence (in p) of pseudo-Lipschitz functions $\{\phi_p\}_{p \in \mathbb{N}_{>0}}$ is* uniformly pseudo-Lipschitz *of order $k$ if, denoting by $L_p$ the pseudo-Lipschitz constant of $\phi_p$, $L_p < \infty$ for each $p$ and $\limsup_{p \to \infty} L_p < \infty$.*

**Theorem 3.** *Under assumptions (A1) - (A5), for any uniformly pseudo-Lipschitz sequence of functions $\psi_p : \mathbb{R}^p \times \mathbb{R}^p \to \mathbb{R}$ and for $\boldsymbol{Z} \sim \mathcal{N}(0, \mathbb{I}_p)$,*

$$\operatorname*{plim}_{p} \psi_p(\widehat{\boldsymbol{\beta}}, \boldsymbol{\beta}) = \lim_{t} \operatorname*{plim}_{p} \mathbb{E}_{\boldsymbol{Z}}[\psi_p(\operatorname{prox}_{J_{\boldsymbol{\alpha}(p)\tau_t}}(\boldsymbol{\beta} + \tau_t \boldsymbol{Z}), \boldsymbol{\beta})],$$

*where $\tau_t$ is defined in* (2.4) *and the expectation is taken with respect to $\boldsymbol{Z}$.*

Theorem 3 tells us that under uniformly pseudo-Lipschitz loss, in the large system limit, distributionally the SLOPE optimizer acts as a 'denoised' version of the truth corrupted by additive Gaussian noise where the denoising function is given by the proximal operator, i.e. within uniformly pseudo-Lipschitz loss $\widehat{\boldsymbol{\beta}}$ can be replaced with $\operatorname{prox}_{J_{\boldsymbol{\alpha}(p)\tau_t}}(\boldsymbol{\beta} + \tau_t \boldsymbol{Z})$ for large $p, t$.

We note that the result [21, Theorem 1] follows by Theorem 3 and their separability result [21, Proposition 1]. To see this, in Theorem 3 consider a special case where $\psi_p(\boldsymbol{x}, \boldsymbol{y}) = \frac{1}{p} \sum \psi(x_i, y_i)$

for function $\psi : \mathbb{R} \times \mathbb{R} \to \mathbb{R}$ that is pseudo-Lipschitz of order $k = 2$. Then it is easy to show that $\psi_p(\cdot, \cdot)$ is uniformly pseudo-Lipschitz of order $k = 2$. The result of Theorem 3 then says that

$$\text{plim}_p \frac{1}{p} \sum_{i=1}^{p} \psi(\widehat{\beta}_i, \beta_i) = \lim_t \text{plim}_p \frac{1}{p} \sum_{i=1}^{p} \mathbb{E}_{\boldsymbol{Z}}[\psi([\text{prox}_{J_{\boldsymbol{\alpha}(p)\tau_t}}(\boldsymbol{\beta} + \tau_t \boldsymbol{Z})]_i, \beta_i)].$$

Then by [21, Proposition 1], restated in Lemma E.1, which says that the proximal operator becomes asymptotically separable as $p \to \infty$, the result of [21, Theorem 1] follows by the Law of Large Numbers and Theorem 1. Namely, for some limiting scalar function $h^t$,

$$\lim_t \text{plim}_p \frac{1}{p} \sum_{i=1}^{p} \mathbb{E}_{\boldsymbol{Z}}[\psi([\text{prox}_{J_{\boldsymbol{\alpha}(p)\tau_t}}(\boldsymbol{\beta} + \tau_t \boldsymbol{Z})]_i, \beta_i)] \overset{(a)}{=} \lim_t \text{plim}_p \frac{1}{p} \sum_{i=1}^{p} \mathbb{E}_{\boldsymbol{Z}}[\psi(h^t([\boldsymbol{\beta} + \tau_t \boldsymbol{Z}]_i), \beta_i)]$$

$$= \lim_t \mathbb{E}_{Z,B}[\psi(h^t(B + \tau_t Z), B)] = \mathbb{E}_{Z,B}[\psi(h^t(B + \tau_* Z), B)].$$

We note in step $(a)$ above, we apply Lemma E.1, using that $\boldsymbol{\alpha}(p)\tau_t$ has an empirical distribution that converges weakly to $A\tau_t$ for $A$ defined by (2.10). The rigorous argument for justifying step $(a)$ by Lemma E.1 requires a bit more technical detail. We give such a rigorous argument, for a similar but different limiting operation, in Appendix D for proving limiting properties of the prox operator (namely, property **(P2)** stated in Appendix B).

We highlight that our Theorem 3 allows the consideration of a non-asymptotic case in $t$. While Theorem 1 motivates an algorithmic way to find a value $\tau_t(p)$ which approximates $\tau_*(p)$ well, Theorem 3 guarantees the accuracy of such approximation for use in practice. One particular use of Theorem 3 is to design the optimal sequence $\boldsymbol{\lambda}$ that achieves the minimum $\tau_*$ and equivalently minimum error [21], though a concrete algorithm for doing so is still under investigation.

We prove Theorem 3 in Appendix B. We show that Theorem 3 follows from Theorem 2 and Lemma B.1, which demonstrates that the state evolution given in (2.4) characterizes the performance of the SLOPE AMP (1.3) via pseudo-Lipschitz loss functions. Finally we show how we use Theorem 3 to study the asymptotic mean-square error between the SLOPE estimator and the truth.

**Corollary 3.2.** *Under assumptions (A1) − (A5),* $\text{plim}_p \|\widehat{\boldsymbol{\beta}} - \boldsymbol{\beta}\|^2 / p = \delta(\tau_*^2 - \sigma_w^2)$.

*Proof.* Applying Theorem 3 to the pseudo-Lipschitz loss function $\psi^1 : \mathbb{R}^p \times \mathbb{R}^p \to \mathbb{R}$, defined as $\psi^1(\boldsymbol{x}, \boldsymbol{y}) = \|\boldsymbol{x} - \boldsymbol{y}\|^2 / p$, we find $\text{plim}_p \frac{1}{p} \|\widehat{\boldsymbol{\beta}} - \boldsymbol{\beta}\|^2 = \lim_t \text{plim}_p \frac{1}{p} \mathbb{E}_{\boldsymbol{Z}}[\|\text{prox}_{J_{\boldsymbol{\alpha}\tau_t}}(\boldsymbol{\beta} + \tau_t \boldsymbol{Z}) - \boldsymbol{\beta}\|^2]$. The desired result follows since $\lim_t \text{plim}_p \frac{1}{p} \mathbb{E}_{\boldsymbol{Z}}[\|\text{prox}_{J_{\boldsymbol{\alpha}\tau_t}}(\boldsymbol{\beta} + \tau_t \boldsymbol{Z}) - \boldsymbol{\beta}\|^2] = \delta(\tau_*^2 - \sigma_w^2)$. To see this, note that $\lim_t \delta(\tau_{t+1}^2 - \sigma_w^2) = \delta(\tau_*^2 - \sigma_w^2)$ and

$$\text{plim}_p \frac{1}{p} \mathbb{E}_{\boldsymbol{Z}}[\|\text{prox}_{J_{\boldsymbol{\alpha}\tau_t}}(\boldsymbol{\beta} + \tau_t \boldsymbol{Z}) - \boldsymbol{\beta}\|^2] = \lim_p \frac{1}{p} \mathbb{E}_{\boldsymbol{Z},\boldsymbol{B}}[\|\text{prox}_{J_{\boldsymbol{\alpha}\tau_t}}(\boldsymbol{B} + \tau_t \boldsymbol{Z}) - \boldsymbol{B}\|^2] = \delta(\tau_{t+1}^2 - \sigma_w^2),$$

for $\boldsymbol{B}$ elementwise i.i.d. $B$ independent of $\boldsymbol{Z} \sim \mathcal{N}(0, \mathbb{I}_p)$. A rigorous argument for the above follows similarly to that used to prove property **(P2)** stated in Appendix B and proved in Appendix D. $\square$

## 4 Discussion and Future Work

This work develops and analyzes the dynamics of an approximate message passing (AMP) algorithm with the purpose of solving the SLOPE convex optimization procedure for high-dimensional linear regression. By employing recent theoretical analysis of AMP when the non-linearities used in the algorithm are non-separable [8], as is the case for the SLOPE problem, we provide a rigorous proof that the proposed AMP algorithm finds the SLOPE solution asymptotically. Moreover empirical evidence suggests that the AMP estimate is already very close to the SLOPE solution even in few iterations. By leveraging our analysis showing that AMP provably solves SLOPE, we provide an exact asymptotic characterization of the $\ell_2$ risk of the SLOPE estimator from the underlying truth and insight into other statistical properties of the SLOPE estimator. Though this asymptotic analysis of the SLOPE solution has been demonstrated in other recent work [21] using a different proof strategy, we have a clear, rigorous statement of where it applies. That is, the analysis in [21] applies *if* the state evolution has a unique fixed point, whereas our Theorem 1 states precise conditions under which this is true. Moreover, we believe that our algorithmic approach offers a more concrete connection between the finite-sample behavior of the SLOPE estimator and its asymptotic distribution.

We now briefly discuss some potential improvements and directions for future research.

**i.i.d. Gaussian measurement matrix assumption.** A limitation of vanilla AMP is that the theory assumes an i.i.d. Gaussian measurement matrix, and moreover, the AMP algorithm can become unstable when the measurement matrix is far from i.i.d., creating the need for heuristic techniques to provide convergence in applications where the measurement matrix is generated by nature (i.e., a real-world experiment or observational study). While, in general, AMP theory provides performance guarantees only for i.i.d. sub-Gaussian data [2, 5], in practice, favorable performance of AMP seems to be more universal. For example, in Fig. 3a, we illustrate the performance of AMP for i.i.d. zero mean, $1/n$ variance design matrices that are *not* Gaussian (one i.i.d. $\pm1$ Bernoulli (top) and one i.i.d. shifted exponential (bottom)). In particular, we note that the exponential prior is *not* sub-Gaussian, so the performance here is not supported by theory. In both cases, AMP converges very fast, thus demonstrating its robustness to distributional assumptions.

On the theoretical side, recent work proposes a variant of AMP, called vector-AMP or VAMP [28], which is a computationally-efficient algorithm that *provably* works for a wide range of design matrices, namely, those that are right rotationally-invariant. For example, [23] studies VAMP for a similar setting as SLOPE. However, the type of nonseparability considered in this work requires the penalty to be separable on subsets of an affine transformation of its input. As such, the setting does not directly apply to SLOPE. To address this, we have built a hybrid, 'SLOPE VAMP', based on code generously shared by the authors of the referenced work [23], which performs very well in the (non-) i.i.d. (non-) Gaussian regime (see Fig. 3a and 3b). Motivated by these promising empirical results, we feel that theoretically understanding SLOPE dynamics with VAMP is an exciting direction for future work.

**Known signal prior assumption.** There is a possibility that, by using EM- or SURE-based AMP strategies, one can remove the known signal prior assumption. Developing such strategies alongside our SLOPE VAMP would provide a quite general framework for recovery of the SLOPE estimator.

**Comparison to 'Bayes-AMP'.** In general, the (statistical) motivation for using methods like LASSO or SLOPE is to perform variable selection, and in addition, for SLOPE, to control the false discovery rate. Both methods are therefore biased and, consequently, 'Bayes-AMP' strategies that are

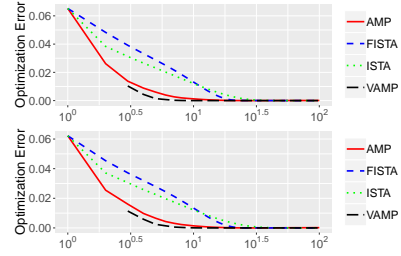

**(a)** i.i.d. $\pm1$ Bernoulli design matrix (top) and i.i.d. shifted exponential design matrix (bottom)

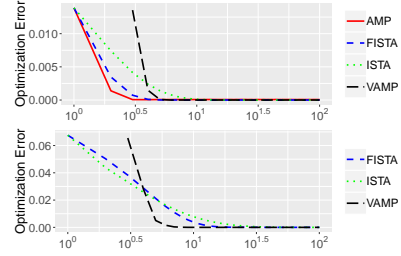

**(b)** i.i.d. Gaussian design matrix (top) and non-i.i.d. right rotationally-invariant design matrix where AMP diverges (bottom)

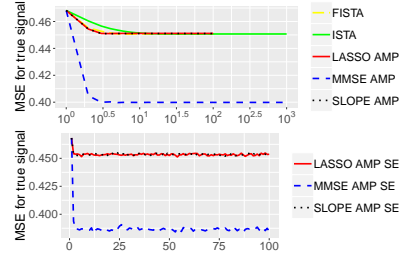

**(c)** i.i.d. Gaussian design matrix

**Figure 3:** Performance of AMP variants in different settings with Bernoulli-Gaussian prior, dimension = 1000, and sample size = 300.

designed to be optimal in terms of MSE will outperform *if performance is based on MSE*. In particular, [14] proves that 'Bayes-AMP' *always* has smaller MSE than that of methods employing convex regularization for a wide class of convex penalties and Gaussian design. Nevertheless, Fig. 3c suggests that SLOPE AMP has MSE that is not too much worse than MMSE AMP.

**Sampling regime.** The asymptotical regime studied here, $n/p \to \delta \in (0, \infty)$, requires that the number of columns of the measurement matrix $p$ grow at the same rate as the number of rows $n$. It is of practical interest to extend the results to high-dimensional settings where $p$ grows faster than $n$.

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
