[Supplementary Material]

# A  Preliminaries on SLOPE

Sorted $\ell_1$-norm penalized regression, henceforth referred to as SLOPE, is a method for recovering $\boldsymbol{\beta}$ when one has knowledge of $\boldsymbol{y}, \boldsymbol{X}$ and the prior distribution $B$. We define the *sorted $\ell_1$-norm* of a vector $\boldsymbol{b} \in \mathbb{R}^p$ with respect to parameter vector $\boldsymbol{\lambda} \in \mathbb{R}^p$ as follows:

$$J_{\boldsymbol{\lambda}}(\boldsymbol{b}) := \sum_{i=1}^{p} \lambda_i |\boldsymbol{b}|_{(i)} \tag{A.1}$$

where $\lambda_1 \geq \ldots \geq \lambda_p \geq 0$ and $|\boldsymbol{b}|_{(1)} \geq \ldots \geq |\boldsymbol{b}|_{(p)}$ are the order statistics of the entries of vector $|\boldsymbol{b}|$.

In general, SLOPE is the solution to the following convex optimization problem

$$\hat{\boldsymbol{\beta}} := \arg\min_{\boldsymbol{b}} \mathcal{C}(\boldsymbol{b}), \qquad \text{where} \qquad \mathcal{C}(\boldsymbol{b}) := \frac{1}{2}\|\boldsymbol{y} - \boldsymbol{X}\boldsymbol{b}\|^2 + J_{\boldsymbol{\lambda}}(\boldsymbol{b}). \tag{A.2}$$

We refer to $\hat{\boldsymbol{\beta}}$ as the SLOPE estimate and $\mathcal{C}(\cdot)$ as the SLOPE cost function. Notice that when $\lambda_1 = \ldots = \lambda_p$, the problem reduces to LASSO since in this case the *sorted $\ell_1$ norm*, $J_{\boldsymbol{\lambda}}(\boldsymbol{b})$, equals the usual $\ell_1$ norm, $\|\boldsymbol{b}\|_1 = \sum_{i=1}^{p} |b_i|$. We refer to the function $\mathcal{C}(\cdot)$ stated in (A.2) as the SLOPE cost function and the SLOPE estimator $\hat{\boldsymbol{\beta}}$ is the one that minimizes the SLOPE cost. We note that the SLOPE cost function $\mathcal{C}(\cdot)$ depends on both $\boldsymbol{y}$ and $\boldsymbol{\lambda}$, so technically a notation like $\mathcal{C}_{(\boldsymbol{y}, \boldsymbol{\lambda})}(\cdot)$ would be more rigorous, however, we don't think that dropping the explicit dependence on $(\boldsymbol{y}, \boldsymbol{\lambda})$ will cause any confusion.

In studying the SLOPE estimator $\hat{\boldsymbol{\beta}}$, it will be useful to study the proximal operator, $\text{prox}_f : \mathbb{R}^p \to \mathbb{R}^p$, associated with a convex function $f : \mathbb{R}^p \to \mathbb{R}$, which is defined for $\boldsymbol{v} \in \mathbb{R}^p$ as

$$\text{prox}_f(\boldsymbol{v}) := \arg\min_{\boldsymbol{b} \in \mathbb{R}^p} \left\{ \frac{1}{2}\|\boldsymbol{v} - \boldsymbol{b}\|^2 + f(\boldsymbol{b}) \right\}. \tag{A.3}$$

For a convex function $f : \mathbb{R}^p \to \mathbb{R}$, we denote the subgradient of $f$ at a point $\boldsymbol{x} \in \mathbb{R}^p$ as $\partial f(\boldsymbol{x})$. We use the following fact relating the proximal operator to the subgradient of the SLOPE norm (A.1) repeatedly throughout the work.

**Fact A.1.** *If* $\text{prox}_{J_{\boldsymbol{\lambda}}}(\boldsymbol{v}_1) = \boldsymbol{v}_2$, *then* $\boldsymbol{v}_1 - \boldsymbol{v}_2 \in \partial J_{\boldsymbol{\lambda}}(\boldsymbol{v}_2)$.

We now describe explicitly $\partial J_{\boldsymbol{\lambda}} \subset \mathbb{R}^p$. Define $\hat{\Pi}_{\boldsymbol{x}} : \mathbb{R}^p \to \{\text{maximal atoms}\}$. In words, $\hat{\Pi}_{\boldsymbol{x}}$ finds the maximal atoms of ranking of the absolute values of $\boldsymbol{x}$. For example if $\boldsymbol{x} = (5, 2, -3, -5)$, then $\hat{\Pi}_{\boldsymbol{x}}$ corresponds to the mapping

$$\begin{pmatrix} 1 & 2 & 3 & 4 \\ \{1,2\} & 4 & 3 & \{1,2\} \end{pmatrix}$$

with $\hat{\Pi}_{\boldsymbol{x}}(\boldsymbol{x}) = (\{5, -5\}, \{5, -5\}, -3, 2)$ and $\hat{\Pi}_{\boldsymbol{x}}^{-1}(\boldsymbol{\lambda}) = (\{\lambda_1, \lambda_2\}, \lambda_4, \lambda_3, \{\lambda_1, \lambda_2\})$.

Define an equivalence relation $x \sim y$ if $|x| = |y|$. Then $\hat{\Pi}_{\boldsymbol{x}}$ partitions elements in $\boldsymbol{x}$ into different equivalence classes $I$. The motivation of using equivalence classes roots from AMP. In calibration, we need $\nabla$ prox which equals the number of non-zero equivalence classes.

**Fact A.2.**

$$\partial J_{\boldsymbol{\lambda}}(\boldsymbol{s}) = \left\{ \boldsymbol{v} \in \mathbb{R}^p : \text{ for each equivalent class } I, \begin{cases} \text{if } \boldsymbol{s}_I \neq 0 \implies \boldsymbol{v}_I \in S([\hat{\Pi}_{\boldsymbol{s}}^{-1}(\boldsymbol{\lambda})]_I) \, \text{sign}(\boldsymbol{s}_I); \\ \text{if } \boldsymbol{s}_I = 0 \implies |\boldsymbol{v}_I| \in S_0([\hat{\Pi}_{\boldsymbol{s}}^{-1}(\boldsymbol{\lambda})]_I) \end{cases} \right\}.$$

Here $S, S_0$ are polytope-related mappings,

$$S(\boldsymbol{u}) := \{\boldsymbol{y} : \boldsymbol{y} = \boldsymbol{A}\boldsymbol{u} \text{ for some doubly stochastic matrix } \boldsymbol{A}\}$$
$$S_0(\boldsymbol{u}) := \{\boldsymbol{y} : \boldsymbol{y} = \boldsymbol{A}\boldsymbol{u} \text{ for some doubly sub-stochastic matrix } \boldsymbol{A}\}$$

where a doubly sub-stochastic matrix is defined as a square matrix of non-negative real numbers, each of whose rows and columns sum to at most 1.

To see why $\partial J_{\boldsymbol{\lambda}}(\boldsymbol{s})$ takes that form in $S$ when $\boldsymbol{s}_I \neq 0$, we look at indices in the same equivalence class: $\boldsymbol{s}_I$ has the same absolute value so the penalties are shared among them. This naturally leads

to an assignment problem: assign jobs (penalties) to workers ($s_i$) where doubly stochastic matrix is commonly seen. For a rigorous proof, we refer to [29] Exercise 8.31. In other words, $S(u)$ is a permutohedron, a convex hull with vertices corresponding to permuted entries of $u$.

On the other hand, $S_0$ does not require that the sharing of penalties and entries is strict: row and/or column sums can be smaller than one. Such difference roots from the partial derivative of $\ell_1$ norm: i.e. $\left| \partial |x| \right| = 1$ when $x \neq 0$ and $\left| \partial |x| \right| \in [0, 1]$ when $x = 0$.

# B    Proof of the Main Theoretical Results

To prove Theorem 3, we use a result guaranteeing that the state evolution given in (2.4) characterizes the performance of the SLOPE AMP algorithm (1.3b), given in Lemma B.1 below. Specifically, Lemma B.1 relates the state evolution (2.4) to the output of the AMP iteration (1.3b) for pseudo-Lipschitz loss functions. This result follows from [8, Theorem 14], which is a general result relating state evolutions to AMP algorithm with non-separable denoisers. In order to apply [8, Theorem 14], we need to demonstrate that our denoiser, i.e. the proximal operator $\text{prox}_{J_{\alpha \tau_t}}(\cdot)$ defined in (A.3), satisfies two additional properties labeled **(P1)** and **(P2)** below.

Define a sequence of denoisers $\{\eta_p^t\}_{p \in \mathbb{N}_{>0}}$ where $\eta_p^t : \mathbb{R}^p \to \mathbb{R}^p$ to be those that apply the proximal operator $\text{prox}_{J_{\alpha \tau_t}}(\cdot)$ defined in (A.3), i.e. for a vector $\boldsymbol{v} \in \mathbb{R}^p$, define

$$\eta_p^t(\boldsymbol{v}) := \text{prox}_{J_{\alpha \tau_t}}(\boldsymbol{v}). \tag{B.1}$$

**(P1)**  For each $t$, denoisers $\eta_p^t(\cdot)$ defined in (B.1) are uniformly Lipschitz (i.e. uniformly pseudo-Lipschitz of order $k = 1$) per Definition 3.1.

**(P2)**  For any $s, t$ with $(\boldsymbol{Z}, \boldsymbol{Z}')$ a pair of length-$p$ vectors such that $(Z_i, Z_i')$ are i.i.d. $\sim \mathcal{N}(0, \boldsymbol{\Sigma})$ for $i \in \{1, 2, \ldots, p\}$ where $\boldsymbol{\Sigma}$ is any $2 \times 2$ covariance matrix, the following limits exist and are finite.

$$\plim_{p \to \infty} \frac{1}{p} \|\boldsymbol{\beta}\|, \quad \plim_{p \to \infty} \frac{1}{p} \mathbb{E}[\boldsymbol{\beta}^\top \eta_p^t(\boldsymbol{\beta} + \boldsymbol{Z})], \quad \text{and} \quad \plim_{p \to \infty} \frac{1}{p} \mathbb{E}[\eta_p^s(\boldsymbol{\beta} + \boldsymbol{Z}')^\top \eta_p^t(\boldsymbol{\beta} + \boldsymbol{Z})].$$

We will show that properties **(P1)** and **(P2)** are satisfied for our problem in Appendix D.

**Lemma B.1.** *[8, Theorem 14] Under assumptions (A1) - (A4), given that (P1) and (P2) are satisfied, for the AMP algorithm in* (1.3b) *and for any uniformly pseudo-Lipschitz sequence of functions* $\phi_n : \mathbb{R}^n \times \mathbb{R}^n \to \mathbb{R}$ *and* $\psi_p : \mathbb{R}^p \times \mathbb{R}^p \to \mathbb{R}$, *let* $\boldsymbol{Z} \sim \mathcal{N}(0, \mathbb{I}_n)$ *and* $\boldsymbol{Z}' \sim \mathcal{N}(0, \mathbb{I}_p)$, *then*

$$\plim_n \left( \phi_n(\boldsymbol{z}^t, \boldsymbol{w}) - \mathbb{E}[\phi_n(\boldsymbol{w} + \sqrt{\tau_t^2 - \sigma^2} \boldsymbol{Z}, \boldsymbol{w})] \right) = 0,$$

$$\plim_p \left( \psi_p(\boldsymbol{\beta}^t + \boldsymbol{X}^\top \boldsymbol{z}^t, \boldsymbol{\beta}) - \mathbb{E}[\psi_p(\boldsymbol{\beta} + \tau_t \boldsymbol{Z}', \boldsymbol{\beta})] \right) = 0,$$

*where* $\tau_t$ *is defined in* (2.4).

We now show that Theorem 3 follows from Lemma B.1 and Theorem 2.

*Proof of Theorem 3.*  First, for any fixed $n$ and $t$, the following bound uses that $\psi_n$ is uniformly pseudo-Lipschitz of order $k$ and the Triangle Inequality,

$$\left| \psi_p(\boldsymbol{\beta}^t, \boldsymbol{\beta}) - \psi_p(\widehat{\boldsymbol{\beta}}, \boldsymbol{\beta}) \right| \leq L \left( 1 + \left( \frac{\|(\boldsymbol{\beta}^t, \boldsymbol{\beta})\|}{\sqrt{2p}} \right)^{k-1} + \left( \frac{\|(\widehat{\boldsymbol{\beta}}, \boldsymbol{\beta})\|}{\sqrt{2p}} \right)^{k-1} \right) \frac{1}{\sqrt{2p}} \|\boldsymbol{\beta}^t - \widehat{\boldsymbol{\beta}}\|$$

$$\leq L \left( 1 + \left( \frac{\|\boldsymbol{\beta}^t\|}{\sqrt{2p}} \right)^{k-1} + \left( \frac{\|\widehat{\boldsymbol{\beta}}\|}{\sqrt{2p}} \right)^{k-1} + \left( \frac{\|\boldsymbol{\beta}\|}{\sqrt{2p}} \right)^{k-1} \right) \frac{1}{\sqrt{2p}} \|\boldsymbol{\beta}^t - \widehat{\boldsymbol{\beta}}\|.$$

Now we take limits on either side of the above, first with respect to $p$ and then with respect to $t$. We note that the term $\frac{1}{\sqrt{n}} \|\boldsymbol{\beta}^t - \widehat{\boldsymbol{\beta}}\|$ vanishes by Theorem 2. Then as long as

$$\lim_t \plim_p \left( \|\boldsymbol{\beta}^t\| / \sqrt{p} \right)^{k-1}, \qquad \plim_p \left( \|\widehat{\boldsymbol{\beta}}\| / \sqrt{p} \right)^{k-1}, \qquad \text{and} \qquad \plim_p \left( \|\boldsymbol{\beta}\| / \sqrt{p} \right)^{k-1}, \tag{B.2}$$

are all finite, we have that

$$\operatorname*{plim}_{p} \psi_p(\widehat{\boldsymbol{\beta}}, \boldsymbol{\beta}) = \lim_t \operatorname*{plim}_{p} \psi_p(\boldsymbol{\beta}^t, \boldsymbol{\beta}).$$

But by Theorem B.1 we also know that

$$\lim_t \operatorname*{plim}_{p} \psi_p(\boldsymbol{\beta}^t, \boldsymbol{\beta}) = \lim_t \operatorname*{plim}_{p} \mathbb{E}[\psi_p(\eta^t(\boldsymbol{\beta} + \tau_t \boldsymbol{Z}), \boldsymbol{\beta})],$$

giving the desired result.

Finally we convince ourself that the limits in (B.2) are finite. Since $k$ finite, that the third term in (B.2) is finite follows by property **(P2)**. The bound for the first term in (B.2) follows by Theorem B.1: since

$$\boldsymbol{\beta}^{t+1} = \operatorname{prox}_{J_{\alpha \tau_t}}(\boldsymbol{X}^\top \boldsymbol{z}^t + \boldsymbol{\beta}^t) = \eta_p^t(\boldsymbol{X}^\top \boldsymbol{z}^t + \boldsymbol{\beta}^t),$$

we apply Theorem B.1 with uniformly pseudo-Lipschitz function $\psi_p(\boldsymbol{\beta}^t + \boldsymbol{X}^\top \boldsymbol{z}^t, \boldsymbol{\beta}) = \frac{1}{p} \|\eta_p^t(\boldsymbol{\beta}^t + \boldsymbol{X}^\top \boldsymbol{z}^t)\|^2$ to get

$$\operatorname*{plim}_{p} \|\boldsymbol{\beta}^t\|^2 / p = \operatorname*{plim}_{p} \mathbb{E}_{\boldsymbol{Z}}[\|\eta_p^t(\boldsymbol{\beta} + \tau_t \boldsymbol{Z})\|^2] / p,$$

for $\boldsymbol{Z} \sim \mathcal{N}(0, \mathbb{I}_p)$. Then we note that by the Lipschitz property of $\eta_p^t$ (property **(P1)**), we have

$$\mathbb{E}[\|\eta_p^t(\boldsymbol{\beta} + \tau_t \boldsymbol{Z})\|^2] \leq \mathbb{E}[\|\boldsymbol{\beta} + \tau_t \boldsymbol{Z}\|^2] \leq 2\|\boldsymbol{\beta}\|^2 + 2p\tau_t^2.$$

Plugging this into the limit result, we find

$$\lim_t \operatorname*{plim}_{p} \|\boldsymbol{\beta}^t\|^2 / p = 2 \operatorname*{plim}_{p} \|\boldsymbol{\beta}\|^2 / p + 2 \lim_t \tau_t^2 = 2\sigma_{\boldsymbol{\beta}}^2 + 2\tau_*^2,$$

where the final inequality follows by Assumption **(A2)** and Property **(P2)**. Then the second term in (B.2) is finite follows by Theorem 3 and the bound on the first term. $\qquad \square$

## C   State Evolution Analysis

*Proof of Theorem 1.* To begin with, we will prove that $\mathsf{F}(\tau^2, \boldsymbol{\alpha}\tau)$ defined in (2.8), namely

$$\mathsf{F}(\tau^2, \boldsymbol{\alpha}\tau) := \sigma^2 + \frac{1}{\delta p} \mathbb{E}\|\operatorname{prox}_{J_{\alpha \tau}}(\boldsymbol{B} + \tau \boldsymbol{Z}) - \boldsymbol{B}\|^2, \tag{C.1}$$

is concave with respect to $\tau^2$. The proof follows along the same lines as the proof of [3, Proposition 1.3], however, whereas the proof of [3, Proposition 1.3] proceeds by explicitly expressing the first derivative of the corresponding function $\mathsf{F}$, and then differentiating on the explicit form to get the second derivative, in SLOPE case, because of the averaging that occurs within the proximal operation, it is extremely difficult to similarly derive an explicit form. To work around this, we keep all differentiation implicit.

First note

$$
\begin{aligned}
\frac{\partial \mathsf{F}}{\partial \tau^2}(\tau^2, \boldsymbol{\alpha}\tau) &= \frac{\partial}{\partial \tau^2} \left( \sigma^2 + \frac{1}{\delta p} \mathbb{E}\|\operatorname{prox}_{J_{\alpha \tau}}(\boldsymbol{B} + \tau \boldsymbol{Z}) - \boldsymbol{B}\|^2 \right) \\
&\overset{(a)}{=} \frac{1}{\delta p} \sum_{i=1}^p \mathbb{E}\left\{ \frac{\partial}{\partial \tau^2} \left( [\operatorname{prox}_{J_{\alpha \tau}}(\boldsymbol{B} + \tau \boldsymbol{Z})]_i - B_i \right)^2 \right\} \\
&= \frac{2}{\delta p} \sum_{i=1}^p \mathbb{E}\left\{ \left( [\operatorname{prox}_{J_{\alpha \tau}}(\boldsymbol{B} + \tau \boldsymbol{Z})]_i - B_i \right) \frac{\partial}{\partial \tau^2} [\operatorname{prox}_{J_{\alpha \tau}}(\boldsymbol{B} + \tau \boldsymbol{Z})]_i \right\}.
\end{aligned}
\tag{C.2}
$$

We note that the interchange between the derivative (a limit) and the expectation in step $(a)$ of the above holds due to a dominated convergence argument that relies on the following lemma.

**Lemma C.1.** *Define an equivalence classes $I_i$ for each index $i = \{1, 2, \ldots, p\}$, defined as*

$$I_i := \{j : |[\text{prox}_{J_{\alpha\tau}}(\boldsymbol{B} + \tau\boldsymbol{Z})]_j| = |[\text{prox}_{J_{\alpha\tau}}(\boldsymbol{B} + \tau\boldsymbol{Z})]_i|\}.$$

*With the above definition, for any $j \in I_i$ we have that $I_j = I_i$. Let $I$ indicate the collection of unique equivalence classes. Then*

$$\left| \frac{\partial}{\partial\tau^2} \frac{1}{p} \|\text{prox}_{J_{\alpha\tau}}(\boldsymbol{B} + \tau\boldsymbol{Z}) - \boldsymbol{B}\|^2 \right| \leq \frac{1}{p} \sum_{I_k \in I} \frac{1}{|I_k|} \Big( \sum_{i \in I_k} |\text{sign}(B_i + \tau Z_i)Z_i - \alpha_i| \Big)^2. \tag{C.3}$$

Lemma C.1 will be proved below, after we solve $\frac{\partial}{\partial\tau^2}[\text{prox}_{J_{\alpha\tau}}(\boldsymbol{B} + \tau\boldsymbol{Z})]_i$.

Now we describe how the bound in Lemma C.1 can be used to produce the dominated convergence result needed in step $(a)$ of (C.2). First note,

$$\frac{1}{p}\mathbb{E}\Big\{ \sum_{I_k \in I} \frac{1}{|I_k|} \Big( \sum_{i \in I_k} |\text{sign}(B_i + \tau Z_i)Z_i - \alpha_i| \Big)^2 \Big\} \leq \frac{2}{p}\mathbb{E}\Big\{ \sum_{I_k \in I} \frac{1}{|I_k|} \Big( \sum_{i \in I_k} \sqrt{Z_i^2 + \alpha_i^2} \Big)^2 \Big\}$$

$$\leq \frac{2}{p}\mathbb{E}\Big\{ \sum_{I_k \in I}\sum_{i \in I_k} (Z_i^2 + \alpha_i^2) \Big\} = \frac{2}{p}\mathbb{E}\Big\{ \sum_{i \in [p]} (Z_i^2 + \alpha_i^2) \Big\}$$

$$= 2 + 2\|\boldsymbol{\alpha}\|^2/p < \infty$$

The first and second inequalities follow from $(\sum_{i=1}^n x_i)^2 \leq n\sum_i x_i^2$. The last inequality comes from entries of $\boldsymbol{\alpha}$ being finite and then $\|\boldsymbol{\alpha}\|^2/p \leq \max_i \alpha_i^2 < \infty$. Therefore we can invoke the dominated convergence theorem that allows the exchange of the derivative and expectation in step $(a)$ of (C.2).

Now we want to further simplify (C.2). For each $1 \leq i \leq p$, we would like to study $\frac{\partial}{\partial\tau^2}[\text{prox}_{J_{\alpha\tau}}(\boldsymbol{B} + \tau\boldsymbol{Z})]_i$. We first note that the mapping $\tau^2 \mapsto [\text{prox}_{J_{\alpha\tau}}(\boldsymbol{B} + \tau\boldsymbol{Z})]_i$ can be considered as $f(g(\tau^2))$, where $g : \mathbb{R} \to \mathbb{R}^{2p}$ is defined as $y \mapsto g(y) := (\boldsymbol{B} + \boldsymbol{Z}\sqrt{y}, \boldsymbol{\alpha}\sqrt{y})$ and $f : \mathbb{R}^{2p} \to \mathbb{R}$ is defined as $(\boldsymbol{a}, \boldsymbol{b}) \mapsto f(\boldsymbol{a}, \boldsymbol{b}) := [\text{prox}_{J_{\boldsymbol{b}}}(\boldsymbol{a})]_i$. Hence,

$$\frac{\partial}{\partial\tau^2}[\text{prox}_{J_{\alpha\tau}}(\boldsymbol{B} + \tau\boldsymbol{Z})]_i = \boldsymbol{J}_{f \circ g}(\tau^2) \stackrel{(a)}{=} \boldsymbol{J}_f(g(\tau^2))\boldsymbol{J}_g(\tau^2)$$

$$= \Big[ \nabla_{\boldsymbol{a}}f(g(\tau^2)), \nabla_{\boldsymbol{b}}f(g(\tau^2)) \Big] \Big[ \frac{\boldsymbol{Z}}{2\tau}, \frac{\boldsymbol{\alpha}}{2\tau} \Big]^\top, \tag{C.4}$$

where $\boldsymbol{J}_h \in \mathbb{R}^{m \times n}$ is the Jacobian matrix of a function $h : \mathbb{R}^n \to \mathbb{R}^m$ and step $(a)$ follows by the chain rule. We denote the proximal operator using a function $\eta : \mathbb{R}^{2p} \to \mathbb{R}^p$ as $\eta(\boldsymbol{a}, \boldsymbol{b}) := \text{prox}_{J_{\boldsymbol{b}}}(\boldsymbol{a})$ and we consider the partial derivatives of $\eta$ with respect to its first and second arguments. We denote

$$\partial_1\eta(\boldsymbol{a}, \boldsymbol{b}) := \text{diag}\Big( \frac{\partial}{\partial a_1}, \frac{\partial}{\partial a_2}, \ldots, \frac{\partial}{\partial a_p} \Big)\eta(\boldsymbol{a}, \boldsymbol{b}), \tag{C.5}$$

and

$$\partial_2\eta(\boldsymbol{a}, \boldsymbol{b}) := \text{diag}\Big( \frac{\partial}{\partial b_1}, \frac{\partial}{\partial b_2}, \ldots, \frac{\partial}{\partial b_p} \Big)\eta(\boldsymbol{a}, \boldsymbol{b}). \tag{C.6}$$

Recall that we have defined the derivatives computed in $\partial_1\eta(\boldsymbol{a}, \boldsymbol{b})$ in (2.2). Note that by anti-symmetry between two arguments, $\frac{d}{db_j}[\eta(\boldsymbol{a}, \boldsymbol{b})]_i = -\text{sign}([\eta(\boldsymbol{a}, \boldsymbol{b})]_j)\frac{d}{da_j}[\eta(\boldsymbol{a}, \boldsymbol{b})]_i$. Then using (2.2), namely the result,

$$\frac{\partial[\text{prox}_{J_\lambda}(\boldsymbol{v})]_i}{\partial v_j} = \frac{\partial[\eta(\boldsymbol{v}, \boldsymbol{\lambda})]_i}{\partial v_j} = \begin{cases} \frac{\text{sign}([\eta(\boldsymbol{v},\boldsymbol{\lambda})]_i) \cdot \text{sign}([\eta(\boldsymbol{v},\boldsymbol{\lambda})]_j)}{\#\{1 \leq k \leq p : |[\eta(\boldsymbol{v},\boldsymbol{\lambda})]_k| = |[\eta(\boldsymbol{v},\boldsymbol{\lambda})]_j|\}}, & \text{if } |[\eta(\boldsymbol{v},\boldsymbol{\lambda})]_j| = |[\eta(\boldsymbol{v},\boldsymbol{\lambda})]_i|, \\ 0, & \text{otherwise}, \end{cases}$$

$$= \frac{\mathbb{I}\{|[\eta(\boldsymbol{v},\boldsymbol{\lambda})]_i| = |[\eta(\boldsymbol{v},\boldsymbol{\lambda})]_j|\}\,\text{sign}([\eta(\boldsymbol{v},\boldsymbol{\lambda})]_i[\eta(\boldsymbol{v},\boldsymbol{\lambda})]_j)}{\#\{1 \leq k \leq p : |[\eta(\boldsymbol{v},\boldsymbol{\lambda})]_k| = |[\eta(\boldsymbol{v},\boldsymbol{\lambda})]_i|\}}$$

we have

$$\frac{d}{da_j}f(\boldsymbol{a}, \boldsymbol{b}) = \frac{d}{da_j}[\eta(\boldsymbol{a}, \boldsymbol{b})]_i = \mathbb{I}\{|[\eta(\boldsymbol{a}, \boldsymbol{b})]_i| = |[\eta(\boldsymbol{a}, \boldsymbol{b})]_j|\}\,\text{sign}([\eta(\boldsymbol{a}, \boldsymbol{b})]_i[\eta(\boldsymbol{a}, \boldsymbol{b})]_j)[\partial_1\eta(\boldsymbol{a}, \boldsymbol{b})]_i,$$

$$\tag{C.7}$$

and similarly,

$$\frac{d}{db_j}f(\boldsymbol{a},\boldsymbol{b}) = \frac{d}{db_j}[\eta(\boldsymbol{a},\boldsymbol{b})]_i = -\mathbb{I}\{|[\eta(\boldsymbol{a},\boldsymbol{b})]_i| = |[\eta(\boldsymbol{a},\boldsymbol{b})]_j|\}\operatorname{sign}([\eta(\boldsymbol{a},\boldsymbol{b})]_i)[\partial_1\eta(\boldsymbol{a},\boldsymbol{b})]_i.$$

Now plugging the above into (C.4), we have

$$\frac{\partial[\operatorname{prox}_{J_{\alpha\tau}}(\boldsymbol{B}+\tau\boldsymbol{Z})]_i}{\partial\tau^2}$$
$$= \frac{1}{2\tau}[\partial_1\eta(\boldsymbol{B}+\tau\boldsymbol{Z},\boldsymbol{\alpha}\tau)]_i\operatorname{sign}([\eta(\boldsymbol{B}+\tau\boldsymbol{Z},\boldsymbol{\alpha}\tau)]_i)\sum_{j\in I_i}(\operatorname{sign}([\eta(\boldsymbol{B}+\tau\boldsymbol{Z},\boldsymbol{\alpha}\tau)]_j)Z_j - \alpha_j)$$

$$(C.8)$$

Then dropping the explicit statement of the $\eta(\cdot,\cdot)$ input to save space, namely writing $\eta_i$ to mean $[\eta(\boldsymbol{B}+\tau\boldsymbol{Z},\boldsymbol{\alpha}\tau)]_i$ or $[\partial_1\eta]_i$ to mean $[\partial_1\eta(\boldsymbol{B}+\tau\boldsymbol{Z},\boldsymbol{\alpha}\tau)]_i$ for example, we plug the result of (C.8) into (C.2) to find

$$\frac{\partial\mathsf{F}}{\partial\tau^2}(\tau^2,\boldsymbol{\alpha}\tau) = \frac{1}{\delta p\tau}\sum_{i=1}^p\sum_{j\in I_i}\mathbb{E}\Big\{(\eta_i - B_i)[\partial_1\eta]_i\operatorname{sign}(\eta_i)(\operatorname{sign}(\eta_j)Z_j - \alpha_j)\Big\}$$

$$= \frac{1}{\delta p}\sum_{i=1}^p\sum_{j\in I_i}\mathbb{E}\{([\partial_1\eta]_i)^2 + (\eta_i - B_i)[\partial_1^2\eta]_i\} - \frac{1}{\delta p\tau}\sum_{i=1}^p\sum_{j\in I_i}\mathbb{E}\Big\{(\eta_i - B_i)[\partial_1\eta]_i\operatorname{sign}(\eta_i)\alpha_j\Big\}.$$

$$(C.9)$$

where the second equality follows by Stein's lemma: for fixed $i$ and $j\in I_i$, for standard Gaussian $Z$ we have $\mathbb{E}\{f(Z)Z\} = \mathbb{E}\{f'(Z)\}$ and therefore,

$$\frac{1}{\tau}\mathbb{E}\{[\partial_1\eta]_i\operatorname{sign}(\eta_i)(\eta_i - B_i)\operatorname{sign}(\eta_j)Z_j\}$$
$$= \mathbb{E}\Big\{\operatorname{sign}(\eta_i)\operatorname{sign}(\eta_j)\Big[(\eta_i - B_i)\frac{d}{da_j}[\partial_1\eta(\boldsymbol{a},\boldsymbol{b})]_i + [\partial_1\eta]_i\frac{d}{da_j}[\eta(\boldsymbol{a},\boldsymbol{b})]_i\Big]\Big\}$$
$$= \mathbb{E}\Big\{(\eta_i - B_i)[\partial_1^2\eta]_i + ([\partial_1\eta]_i)^2\Big\}.$$

where in the last step we use the definition of $\frac{d}{da_j}[\eta(\boldsymbol{a},\boldsymbol{b})]_i$ given in (C.7) and the fact that $\frac{d}{da_j}[\partial_1\eta(\boldsymbol{a},\boldsymbol{b})]_i = \operatorname{sign}(\eta_i)\operatorname{sign}(\eta_j)[\partial_1^2\eta(\boldsymbol{a},\boldsymbol{b})]_i$.

Therefore, simplifying (C.9), we have shown

$$(\delta p\tau)\times\frac{\partial\mathsf{F}}{\partial\tau^2}(\tau^2,\boldsymbol{\alpha}\tau) = \sum_{i=1}^p\mathbb{E}\Big\{\tau|I_i|\Big([\partial_1\eta]_i^2 + (\eta_i - B_i)[\partial_1^2\eta]_i\Big) - [\partial_1\eta]_i\operatorname{sign}(\eta_i)(\eta_i - B_i)\sum_{j\in I_i}\alpha_j\Big\}.$$

$$(C.10)$$

We now have the tools to prove Lemma C.1.

*Proof of Lemma C.1.* First,

$$\frac{\partial}{\partial\tau^2}\frac{1}{p}\|\operatorname{prox}_{J_{\alpha\tau}}(\boldsymbol{B}+\tau\boldsymbol{Z}) - \boldsymbol{B}\|^2 = \frac{1}{p}\sum_{i=1}^p\frac{\partial}{\partial\tau^2}\Big([\operatorname{prox}_{J_{\alpha\tau}}(\boldsymbol{B}+\tau\boldsymbol{Z})]_i - B_i\Big)^2$$
$$= \frac{2}{p}\sum_{i=1}^p\Big([\operatorname{prox}_{J_{\alpha\tau}}(\boldsymbol{B}+\tau\boldsymbol{Z})]_i - B_i\Big)\frac{\partial}{\partial\tau^2}[\operatorname{prox}_{J_{\alpha\tau}}(\boldsymbol{B}+\tau\boldsymbol{Z})]_i.$$

As in the work above, we denote the proximal operator using a function $\eta:\mathbb{R}^{2p}\to\mathbb{R}^p$ as $\eta(\boldsymbol{a},\boldsymbol{b}) := \operatorname{prox}_{J_b}(\boldsymbol{a})$. Now from (C.8), denoting $I_i := \{j : |[\eta(\boldsymbol{a},\boldsymbol{b})]_j| = |[\eta(\boldsymbol{a},\boldsymbol{b})]_i|\}$,

$$\frac{\partial[\operatorname{prox}_{J_{\alpha\tau}}(\boldsymbol{B}+\tau\boldsymbol{Z})]_i}{\partial\tau^2}$$
$$= \frac{1}{2\tau}[\partial_1\eta(\boldsymbol{B}+\tau\boldsymbol{Z},\boldsymbol{\alpha}\tau)]_i\operatorname{sign}([\eta(\boldsymbol{B}+\tau\boldsymbol{Z},\boldsymbol{\alpha}\tau)]_i)\sum_{j\in I_i}(\operatorname{sign}([\eta(\boldsymbol{B}+\tau\boldsymbol{Z},\boldsymbol{\alpha}\tau)]_j)Z_j - \alpha_j).$$

Noting that $\mathrm{sign}([\eta(\boldsymbol{B}+\tau\boldsymbol{Z},\boldsymbol{\alpha}\tau)]_i) = \mathrm{sign}(B_i+\tau Z_i)$ for any index $i$ when $[\eta(\boldsymbol{B}+\tau\boldsymbol{Z},\boldsymbol{\alpha}\tau)]_i \neq 0$, we therefore have,

$$\Big|\frac{\partial}{\partial\tau^2}\frac{1}{p}\|\mathrm{prox}_{J_{\alpha\tau}}(\boldsymbol{B}+\tau\boldsymbol{Z})-\boldsymbol{B}\|^2\Big|$$

$$= \frac{1}{\tau p}\Big|\sum_{i=1}^{p}([\eta(\boldsymbol{B}+\tau\boldsymbol{Z},\boldsymbol{\alpha}\tau)]_i - B_i)\,[\partial_1\eta(\boldsymbol{B}+\tau\boldsymbol{Z},\boldsymbol{\alpha}\tau)]_i\,\mathrm{sign}(B_i+\tau Z_i)\sum_{j\in I_i}(\mathrm{sign}(B_j+\tau Z_j)Z_j-\alpha_j)\Big|.$$

$$\text{(C.11)}$$

Now using the fact that the averaging operation reduces the dot product, meaning informally, that for a vector $\boldsymbol{v}\in\mathbb{R}^p$,

$$(\mathrm{mean}(\boldsymbol{v}),...,\mathrm{mean}(\boldsymbol{v}))\cdot\boldsymbol{v}\le\|\boldsymbol{v}\|^2.$$

Using this in (C.11), we have for any $i\in\{1,2,\dots,p\}$ that $\eta(\boldsymbol{B}+\tau\boldsymbol{Z},\boldsymbol{\alpha}\tau)]_i - B_i$ can be replaced with $B_i + \tau Z_i - \mathrm{sign}(B_i+\tau Z_i)\alpha_i\tau - B_i$. Therefore,

$$\Big|\frac{\partial}{\partial\tau^2}\frac{1}{p}\|\mathrm{prox}_{J_{\alpha\tau}}(\boldsymbol{B}+\tau\boldsymbol{Z})-\boldsymbol{B}\|^2\Big|$$

$$\le \frac{1}{p}\Big|\sum_{i=1}^{p}\sum_{j\in I_i}(Z_i-\mathrm{sign}(B_i+\tau Z_i)\alpha_i)\,[\partial_1\eta(\boldsymbol{B}+\tau\boldsymbol{Z},\boldsymbol{\alpha}\tau)]_i\,\mathrm{sign}(B_i+\tau Z_i)(\mathrm{sign}(B_j+\tau Z_j)Z_j-\alpha_j)\Big|$$

$$= \frac{1}{p}\Big|\sum_{i=1}^{p}\sum_{j\in I_i}(\mathrm{sign}(B_i+\tau Z_i)Z_i-\alpha_i)(\mathrm{sign}(B_j+\tau Z_j)Z_j-\alpha_j)\,[\partial_1\eta(\boldsymbol{B}+\tau\boldsymbol{Z},\boldsymbol{\alpha}\tau)]_i\Big|.$$

$$\text{(C.12)}$$

Next, using that $0\le|[\partial_1\eta]_i|\le 1/|I_i|$,

$$\Big|\sum_{i=1}^{p}\sum_{j\in I_i}(\mathrm{sign}(B_i+\tau Z_i)Z_i-\alpha_i)(\mathrm{sign}(B_j+\tau Z_j)Z_j-\alpha_j)\,[\partial_1\eta(\boldsymbol{B}+\tau\boldsymbol{Z},\boldsymbol{\alpha}\tau)]_i\Big|$$

$$\le \sum_{i=1}^{p}\frac{1}{|I_i|}\sum_{j\in I_i}\Big|(\mathrm{sign}(B_i+\tau Z_i)Z_i-\alpha_i)(\mathrm{sign}(B_j+\tau Z_j)Z_j-\alpha_j)\Big|.$$

Finally we make the following observation. Any equivalence class $I_i$ is a collection of indices $j\in\{1,2,\dots,p\}$ such that $|[\mathrm{prox}_{J_{\alpha\tau}}(\boldsymbol{B}+\tau\boldsymbol{Z})]_j| = |[\mathrm{prox}_{J_{\alpha\tau}}(\boldsymbol{B}+\tau\boldsymbol{Z})]_i|$. Therefore, for any $j\in I_i$ we have that $I_j = I_i$. We then let $I$ indicate the collection of unique equivalence classes, and we have

$$\sum_{i=1}^{p}\frac{1}{|I_i|}\sum_{j\in I_i}\Big|(\mathrm{sign}(B_i+\tau Z_i)Z_i-\alpha_i)(\mathrm{sign}(B_j+\tau Z_j)Z_j-\alpha_j)\Big|$$

$$= \sum_{I_i\in I}\frac{1}{|I_k|}\sum_{i,j\in I_k}\Big|(\mathrm{sign}(B_i+\tau Z_i)Z_i-\alpha_i)(\mathrm{sign}(B_j+\tau Z_j)Z_j-\alpha_j)\Big|.$$

Now plugging back into (C.12),

$$\Big|\frac{\partial}{\partial\tau^2}\frac{1}{p}\|\mathrm{prox}_{J_{\alpha\tau}}(\boldsymbol{B}+\tau\boldsymbol{Z})-\boldsymbol{B}\|^2\Big|$$

$$\le \frac{1}{p}\sum_{I_k\in I}\frac{1}{|I_k|}\sum_{i,j\in I_k}\Big|(\mathrm{sign}(B_i+\tau Z_i)Z_i-\alpha_i)(\mathrm{sign}(B_j+\tau Z_j)Z_j-\alpha_j)\Big|$$

$$= \frac{1}{p}\sum_{I_k\in I}\frac{1}{|I_k|}\Big(\sum_{j\in I_k}|\mathrm{sign}(B_j+\tau Z_j)Z_j-\alpha_j|\Big)^2.$$

$$\square$$

Now considering (C.10), for simplicity in our future calculations, we suppress $|I_i|$ to 1 without loss of generality, since $|I_i|$ is a constant and all operations below preserves linearity. To see this, recall

that $I_i := \{j : |[\eta(\boldsymbol{B} + \tau\boldsymbol{Z}, \boldsymbol{\alpha}\tau)]_j| = |[\eta(\boldsymbol{B} + \tau\boldsymbol{Z}, \boldsymbol{\alpha}\tau)]_i|\}$ and note that when $|[\eta(\boldsymbol{B} + \tau\boldsymbol{Z}, \boldsymbol{\alpha}\tau)]_j|$ equals $|[\eta(\boldsymbol{B} + \tau\boldsymbol{Z}, \boldsymbol{\alpha}\tau)]_i|$, the terms will remain equal after small changes in $\tau$. Note that similarly, $\sum_{j \in I_i} \alpha_j$, will pass through future calculations as a constant. We will convince the readers by generalizing back to multiple-element equivalence classes at the end. Therefore the above becomes

$$
\begin{aligned}
(\delta p \tau) &\times \frac{\partial \mathsf{F}}{\partial \tau^2}(\tau^2, \boldsymbol{\alpha}\tau) \\
&= \sum_{i=1}^{p} \tau \mathbb{E}\Big\{ ([\partial_1 \eta(\boldsymbol{B} + \tau\boldsymbol{Z}, \boldsymbol{\alpha}\tau)]_i)^2 + ([\eta(\boldsymbol{B} + \tau\boldsymbol{Z}, \boldsymbol{\alpha}\tau)]_i - B_i)[\partial_1^2 \eta(\boldsymbol{B} + \tau\boldsymbol{Z}, \boldsymbol{\alpha}\tau)]_i \Big\} \\
&\quad - \sum_{i=1}^{p} \alpha_i \mathbb{E}\Big\{ \operatorname{sign}([\eta(\boldsymbol{B} + \tau\boldsymbol{Z}, \boldsymbol{\alpha}\tau)]_i)([\eta(\boldsymbol{B} + \tau\boldsymbol{Z}, \boldsymbol{\alpha}\tau)]_i - B_i)[\partial_1 \eta(\boldsymbol{B} + \tau\boldsymbol{Z}, \boldsymbol{\alpha}\tau)]_i \Big\}.
\end{aligned}
$$
(C.13)

In what follows we will need to take care with the points $(\boldsymbol{x}, \boldsymbol{y})$ such that $[\partial_1^2 \eta(\boldsymbol{x}, \boldsymbol{y})]_i$ is not equal to 0. We will refer to such points as 'kink' points, since these are points where the partial derivative jumps (and the second partial gradient acts like Dirac delta function $\delta(x)$), or in other words the points where the two (sorted, averaged) arguments in $\eta$ are equal to each other. Informally, we define a 'kink' point as an index where the sorted vector $\boldsymbol{x}$ matches the corresponding threshold $\boldsymbol{y}$ exactly. In LASSO, for example, the correspond to the 'kinks' of the soft-thresholding function. We have

$$
[\partial_1^2 \eta(\boldsymbol{B} + \tau\boldsymbol{Z}, \boldsymbol{\alpha}\tau)]_i = \delta(B_i + \tau Z_i - \alpha_i \tau) - \delta(B_i + \tau Z_i + \alpha_i \tau)
$$

and

$$
\underset{\boldsymbol{Z}, \boldsymbol{B}}{\mathbb{E}} \Big\{ ([\eta(\boldsymbol{B} + \tau\boldsymbol{Z}, \boldsymbol{\alpha}\tau)]_i - B_i)[\partial_1^2 \eta(\boldsymbol{B} + \tau\boldsymbol{Z}, \boldsymbol{\alpha}\tau)]_i \Big\} = -\frac{1}{\tau} \underset{\boldsymbol{B}}{\mathbb{E}} \Big\{ B_i[\phi(\alpha_i - \frac{1}{\tau}B_i) - \phi(-\alpha_i - \frac{1}{\tau}B_i)] \Big\}.
$$

Therefore,

$$
\begin{aligned}
(\delta p \tau) &\times \frac{\partial \mathsf{F}}{\partial \tau^2}(\tau^2, \boldsymbol{\alpha}\tau) \\
&= \tau \mathbb{E}||\partial_1 \eta(\boldsymbol{B} + \tau\boldsymbol{Z}, \boldsymbol{\alpha}\tau)||^2 - \underset{\boldsymbol{B}}{\mathbb{E}} \Big\{ \boldsymbol{B}^\top [\phi(\boldsymbol{\alpha} - \frac{1}{\tau}\boldsymbol{B}) - \phi(-\boldsymbol{\alpha} - \frac{1}{\tau}\boldsymbol{B})] \Big\} \\
&\quad - \mathbb{E}\Big\{ \big[ \boldsymbol{\alpha} \odot \operatorname{sign}(\eta(\boldsymbol{B} + \tau\boldsymbol{Z}, \boldsymbol{\alpha}\tau)) \odot (\eta(\boldsymbol{B} + \tau\boldsymbol{Z}, \boldsymbol{\alpha}\tau) - \boldsymbol{B}) \big]^\top \partial_1 \eta(\boldsymbol{B} + \tau\boldsymbol{Z}, \boldsymbol{\alpha}\tau) \Big\}.
\end{aligned}
$$
(C.14)

Now that we've shown the first derivative, we want to consider the second derivative in order to prove concavity.

Notice, however, that in order to prove concavity of $\mathsf{F}(\tau^2, \boldsymbol{\alpha}\tau)$ it suffices to show $\frac{\partial}{\partial \tau}[\frac{\partial \mathsf{F}}{\partial \tau^2}(\tau^2, \boldsymbol{\alpha}\tau)] \leq 0$ because $\frac{\partial}{\partial \tau^2}(\frac{\partial \mathsf{F}}{\partial \tau^2}) = \frac{\partial \tau}{\partial \tau^2}[\frac{\partial}{\partial \tau}(\frac{\partial \mathsf{F}}{\partial \tau^2})] = \frac{1}{2\tau}[\frac{\partial}{\partial \tau}(\frac{\partial \mathsf{F}}{\partial \tau^2})]$.

We now show $\frac{\partial}{\partial \tau}[\frac{\partial \mathsf{F}}{\partial \tau^2}(\tau^2, \boldsymbol{\alpha}\tau)] \leq 0$. First,

$$
\begin{aligned}
(\delta p) &\times \frac{\partial}{\partial \tau}\Big[ \frac{\partial \mathsf{F}}{\partial \tau^2}(\tau^2, \boldsymbol{\alpha}\tau) \Big] \\
&= \frac{\partial}{\partial \tau}\mathbb{E}||\partial_1 \eta(\boldsymbol{B} + \tau\boldsymbol{Z}, \boldsymbol{\alpha}\tau)||^2 - \frac{\partial}{\partial \tau}\frac{1}{\tau}\mathbb{E}\Big\{ \boldsymbol{B}^\top [\phi(\boldsymbol{\alpha} - \frac{1}{\tau}\boldsymbol{B}) - \phi(-\boldsymbol{\alpha} - \frac{1}{\tau}\boldsymbol{B})] \Big\} \\
&\quad - \frac{\partial}{\partial \tau}\frac{1}{\tau}\mathbb{E}\Big\{ \big[ \boldsymbol{\alpha} \odot \operatorname{sign}(\eta(\boldsymbol{B} + \tau\boldsymbol{Z}, \boldsymbol{\alpha}\tau)) \odot (\eta(\boldsymbol{B} + \tau\boldsymbol{Z}, \boldsymbol{\alpha}\tau) - \boldsymbol{B}) \big]^\top \partial_1 \eta(\boldsymbol{B} + \tau\boldsymbol{Z}, \boldsymbol{\alpha}\tau) \Big\}.
\end{aligned}
$$
(C.15)

Now we apply chain rule to the first term on the right side of (C.15):

$$
\begin{aligned}
\frac{\partial}{\partial \tau}\mathbb{E}||\partial_1 \eta(\boldsymbol{B} + \tau\boldsymbol{Z}, \boldsymbol{\alpha}\tau)||^2 &= \frac{\partial}{\partial \tau}\sum_{i=1}^{p} \mathbb{E}\{([\partial_1 \eta(\boldsymbol{B} + \tau\boldsymbol{Z}, \boldsymbol{\alpha}\tau)]_i)^2\} \\
&= 2\sum_{i=1}^{p} \mathbb{E}\Big\{ [\partial_1 \eta(\boldsymbol{B} + \tau\boldsymbol{Z}, \boldsymbol{\alpha}\tau)]_i \frac{\partial}{\partial \tau}[\partial_1 \eta(\boldsymbol{B} + \tau\boldsymbol{Z}, \boldsymbol{\alpha}\tau)]_i \Big\}.
\end{aligned}
$$
(C.16)

Similarly to the work shown in Lemma C.1 and the description immediately following, a dominated convergence argument justifies the interchange of expectation and differentiation. We note that by (C.8),

$$\frac{\partial}{\partial \tau}[\eta(\boldsymbol{B} + \tau\boldsymbol{Z}, \boldsymbol{\alpha}\tau)]_i = [Z_i - \alpha_i \operatorname{sign}(B_i + \tau Z_i)][\partial_1 \eta(\boldsymbol{B} + \tau\boldsymbol{Z}, \boldsymbol{\alpha}\tau)]_i,$$

and a similar argument gives

$$\begin{aligned}
\frac{\partial}{\partial \tau}[\partial_1 \eta(\boldsymbol{B} + \tau\boldsymbol{Z}, \boldsymbol{\alpha}\tau)]_i &= Z_i[\partial_1^2 \eta(\boldsymbol{B} + \tau\boldsymbol{Z}, \boldsymbol{\alpha}\tau)]_i + \alpha_i[\partial_2 \partial_1 \eta(\boldsymbol{B} + \tau\boldsymbol{Z}, \boldsymbol{\alpha}\tau)]_i \\
&= [Z_i - \alpha_i \operatorname{sign}(B_i + \tau Z_i)][\partial_1^2 \eta(\boldsymbol{B} + \tau\boldsymbol{Z}, \boldsymbol{\alpha}\tau)]_i.
\end{aligned} \tag{C.17}$$

Plugging (C.17) into (C.16),

$$\begin{aligned}
&\frac{\partial}{\partial \tau}\mathbb{E}\|\partial_1 \eta(\boldsymbol{B} + \tau\boldsymbol{Z}, \boldsymbol{\alpha}\tau)\|^2 \\
&= 2\sum_{i=1}^p \mathbb{E}\Big\{[Z_i - \alpha_i \operatorname{sign}(B_i + \tau Z_i)][\partial_1 \eta(\boldsymbol{B} + \tau\boldsymbol{Z}, \boldsymbol{\alpha}\tau)]_i[\partial_1^2 \eta(\boldsymbol{B} + \tau\boldsymbol{Z}, \boldsymbol{\alpha}\tau)]_i\Big\} \\
&= 2\sum_{i=1}^p \mathbb{E}\Big\{[\partial_1 \eta(\boldsymbol{B} + \tau\boldsymbol{Z}, \boldsymbol{\alpha}\tau)]_i[(Z_i - \alpha_i)\delta([\boldsymbol{B} + \tau(\boldsymbol{Z} - \boldsymbol{\alpha})]_i) - (Z_i + \alpha_i)\delta([\boldsymbol{B} + \tau(\boldsymbol{Z} + \boldsymbol{\alpha})]_i)]\Big\},
\end{aligned} \tag{C.18}$$

where the final equality follows since for any index $i$, $[\partial_1^2 \eta(\boldsymbol{x}, \boldsymbol{y})]_i = 0$ at almost every $(\boldsymbol{x}, \boldsymbol{y})$. Namely, the points $(\boldsymbol{x}, \boldsymbol{y})$ such that $[\partial_1^2 \eta(\boldsymbol{x}, \boldsymbol{y})]_i$ is not equal to 0 are the 'kink' points, as described above, where the partial derivative is undefined and the two (sorted, averaged) arguments are equal. Namely, in these cases $\boldsymbol{B} + \tau\boldsymbol{Z} = \boldsymbol{\alpha}\tau$ or $\boldsymbol{B} + \tau\boldsymbol{Z} = -\boldsymbol{\alpha}\tau$. We note that

$$\begin{aligned}
&2\sum_{i=1}^p \mathbb{E}\Big\{[\partial_1 \eta(\boldsymbol{B} + \tau\boldsymbol{Z}, \boldsymbol{\alpha}\tau)]_i[(Z_i - \alpha_i)\delta([\boldsymbol{B} + \tau(\boldsymbol{Z} - \boldsymbol{\alpha})]_i) - (Z_i + \alpha_i)\delta([\boldsymbol{B} + \tau(\boldsymbol{Z} + \boldsymbol{\alpha})]_i)]\Big\} \\
&= 2\mathbb{E}\Big\{[\partial_1 \eta(\boldsymbol{B} + \tau\boldsymbol{Z}, \boldsymbol{\alpha}\tau)]^\top[(\boldsymbol{Z} - \boldsymbol{\alpha}) \odot \delta(\boldsymbol{B} + \tau(\boldsymbol{Z} - \boldsymbol{\alpha})) - (\boldsymbol{Z} + \boldsymbol{\alpha}) \odot \delta(\boldsymbol{B} + \tau(\boldsymbol{Z} + \boldsymbol{\alpha}))]\Big\}.
\end{aligned}$$

Next we see that

$$\begin{aligned}
&(\boldsymbol{Z} - \boldsymbol{\alpha}) \odot \delta(\boldsymbol{B} + \tau(\boldsymbol{Z} - \boldsymbol{\alpha})) - (\boldsymbol{Z} + \boldsymbol{\alpha}) \odot \delta(\boldsymbol{B} + \tau(\boldsymbol{Z} + \boldsymbol{\alpha})) \\
&= -(\boldsymbol{B}/\tau) \odot \Big[\delta(\boldsymbol{B} + \tau(\boldsymbol{Z} - \boldsymbol{\alpha})) - \delta(\boldsymbol{B} + \tau(\boldsymbol{Z} + \boldsymbol{\alpha}))\Big]
\end{aligned}$$

Hence, using $\phi(z) = \frac{1}{\sqrt{2\pi}}\exp(-z^2/2)$ to be the standard Gaussian density,

$$\begin{aligned}
&\frac{\partial}{\partial \tau}\mathbb{E}\|\partial_1 \eta(\boldsymbol{B} + \tau\boldsymbol{Z}, \boldsymbol{\alpha}\tau)\|^2 \\
&= 2\mathbb{E}_{\boldsymbol{B}}\mathbb{E}_{\boldsymbol{Z}|\boldsymbol{B}}\Big\{[\partial_1 \eta(\boldsymbol{b} + \tau\boldsymbol{Z}, \boldsymbol{\alpha}\tau)]^\top[(\boldsymbol{Z} - \boldsymbol{\alpha}) \odot \delta(\boldsymbol{b} + \tau(\boldsymbol{Z} - \boldsymbol{\alpha})) - (\boldsymbol{Z} + \boldsymbol{\alpha}) \odot \delta(\boldsymbol{b} + \tau(\boldsymbol{Z} + \boldsymbol{\alpha}))]|\boldsymbol{B} = \boldsymbol{b}\Big\} \\
&\overset{(a)}{=} -\frac{1}{\tau^2}\mathbb{E}_{\boldsymbol{B}}\Big\{\boldsymbol{b}^\top \phi(\boldsymbol{\alpha} - (\boldsymbol{b}/\tau)) - \boldsymbol{b}^\top \phi(\boldsymbol{\alpha} + (\boldsymbol{b}/\tau))|\boldsymbol{B} = \boldsymbol{b}\Big\} \\
&= -\frac{1}{\tau^2}\mathbb{E}_{\boldsymbol{B}}\Big\{\boldsymbol{B}^\top\Big[\phi(\boldsymbol{\alpha} - (\boldsymbol{B}/\tau)) - \phi(\boldsymbol{\alpha} + (\boldsymbol{B}/\tau))\Big]\Big\}.
\end{aligned} \tag{C.19}$$

In step $(a)$ in the above, the extra $1/\tau$ scaling comes from the fact that $\tau\boldsymbol{Z}$ is the input to the Dirac delta.

Then, using $\phi'(u) = -u\phi(u)$, the second term in (C.15) equals

$$\begin{aligned}
&-\frac{\partial}{\partial \tau}\frac{1}{\tau}\mathbb{E}\Big\{\boldsymbol{B}^\top[\phi(\boldsymbol{\alpha} - \frac{1}{\tau}\boldsymbol{B}) - \phi(-\boldsymbol{\alpha} - \frac{1}{\tau}\boldsymbol{B})]\Big\} \\
&= \frac{1}{\tau^2}\mathbb{E}\Big\{\boldsymbol{B}^\top[\phi(\boldsymbol{\alpha} - \frac{1}{\tau}\boldsymbol{B}) - \phi(-\boldsymbol{\alpha} - \frac{1}{\tau}\boldsymbol{B})]\Big\} \\
&\quad -\frac{1}{\tau^3}\mathbb{E}\Big\{(\boldsymbol{B}^2)^\top[(\frac{1}{\tau}\boldsymbol{B} - \boldsymbol{\alpha}) \odot \phi(\boldsymbol{\alpha} - \frac{1}{\tau}\boldsymbol{B}) - (\frac{1}{\tau}\boldsymbol{B} + \boldsymbol{\alpha}) \odot \phi(-\boldsymbol{\alpha} - \frac{1}{\tau}\boldsymbol{B})]\Big\}
\end{aligned} \tag{C.20}$$

Now we study the third term in (C.15). First, recalling $\partial_1\eta = -\partial_2\eta$ we have

$$\frac{\partial}{\partial\tau}\frac{1}{\tau}\mathbb{E}\Big\{\big[\boldsymbol{\alpha}\odot\text{sign}(\boldsymbol{B}+\tau\boldsymbol{Z})\odot(\eta(\boldsymbol{B}+\tau\boldsymbol{Z},\boldsymbol{\alpha}\tau)-\boldsymbol{B})\big]^\top\partial_1\eta(\boldsymbol{B}+\tau\boldsymbol{Z},\boldsymbol{\alpha}\tau)\Big\}$$

$$=-\frac{1}{\tau^2}\mathbb{E}\Big\{\big[\boldsymbol{\alpha}\odot\text{sign}(\boldsymbol{B}+\tau\boldsymbol{Z})\odot(\eta(\boldsymbol{B}+\tau\boldsymbol{Z},\boldsymbol{\alpha}\tau)-\boldsymbol{B})\big]^\top\partial_1\eta(\boldsymbol{B}+\tau\boldsymbol{Z},\boldsymbol{\alpha}\tau)\Big\} \qquad\text{(C.21)}$$

$$+\frac{1}{\tau}\sum_{i=1}^{p}\alpha_i\frac{\partial}{\partial\tau}\mathbb{E}\Big\{\text{sign}(B_i+\tau Z_i)([\eta(\boldsymbol{B}+\tau\boldsymbol{Z},\boldsymbol{\alpha}\tau)]_i-B_i)[\partial_1\eta(\boldsymbol{B}+\tau\boldsymbol{Z},\boldsymbol{\alpha}\tau)]_i\Big\},$$

and therefore we study $\frac{\partial}{\partial\tau}\mathbb{E}\Big\{\text{sign}(B_i+\tau Z_i)([\eta(\boldsymbol{B}+\tau\boldsymbol{Z},\boldsymbol{\alpha}\tau)]_i-B_i)[\partial_1\eta(\boldsymbol{B}+\tau\boldsymbol{Z},\boldsymbol{\alpha}\tau)]_i\Big\}$. We
agin note that in (C.21), a dominated convergence argument will justify the interchange of derivative
and expectation. This is done similarly to what was studied in Lemma C.1 and the description
immediately following.

By (C.8) and (C.17),

$$\frac{\partial}{\partial\tau}[\eta(\boldsymbol{B}+\tau\boldsymbol{Z},\boldsymbol{\alpha}\tau)]_i=[Z_i-\alpha_i\,\text{sign}(B_i+\tau Z_i)][\partial_1\eta(\boldsymbol{B}+\tau\boldsymbol{Z},\boldsymbol{\alpha}\tau)]_i,$$

and

$$\frac{\partial}{\partial\tau}[\partial_1\eta(\boldsymbol{B}+\tau\boldsymbol{Z},\boldsymbol{\alpha}\tau)]_i=[Z_i-\alpha_i\,\text{sign}(B_i+\tau Z_i)][\partial_1^2\eta(\boldsymbol{B}+\tau\boldsymbol{Z},\boldsymbol{\alpha}\tau)]_i.$$

Then using an argument as in (C.4), we have that

$$-\frac{\partial}{\partial\tau}\mathbb{E}\Big\{\text{sign}(B_i+\tau Z_i)([\eta(\boldsymbol{B}+\tau\boldsymbol{Z},\boldsymbol{\alpha}\tau)]_i-B_i)[\partial_1\eta(\boldsymbol{B}+\tau\boldsymbol{Z},\boldsymbol{\alpha}\tau)]_i\Big\}$$

$$=\mathbb{E}\Big\{[Z_i-\alpha_i\,\text{sign}(B_i+\tau Z_i)]([\partial_1\eta(\boldsymbol{B}+\tau\boldsymbol{Z},\boldsymbol{\alpha}\tau)]_i)^2\Big\} \qquad\text{(C.22)}$$

$$+\mathbb{E}\Big\{[Z_i-\alpha_i\,\text{sign}(B_i+\tau Z_i)]([\eta(\boldsymbol{B}+\tau\boldsymbol{Z},\boldsymbol{\alpha}\tau)]_i-B_i)[\partial_1^2\eta(\boldsymbol{B}+\tau\boldsymbol{Z},\boldsymbol{\alpha}\tau)]_i\Big\}.$$

Plugging (C.22) back into (C.21), we have

$$\frac{\partial}{\partial\tau}\frac{1}{\tau}\mathbb{E}\Big\{\big[\boldsymbol{\alpha}\odot\text{sign}(\boldsymbol{B}+\tau\boldsymbol{Z})\odot(\eta(\boldsymbol{B}+\tau\boldsymbol{Z},\boldsymbol{\alpha}\tau)-\boldsymbol{B})\big]^\top\partial_1\eta(\boldsymbol{B}+\tau\boldsymbol{Z},\boldsymbol{\alpha}\tau)\Big\}$$

$$=-\frac{1}{\tau^2}\mathbb{E}\Big\{\big[\boldsymbol{\alpha}\odot\text{sign}(\boldsymbol{B}+\tau\boldsymbol{Z})\odot(\eta(\boldsymbol{B}+\tau\boldsymbol{Z},\boldsymbol{\alpha}\tau)-\boldsymbol{B})\big]^\top\partial_1\eta(\boldsymbol{B}+\tau\boldsymbol{Z},\boldsymbol{\alpha}\tau)\Big\}$$

$$+\frac{1}{\tau}\mathbb{E}\Big\{\big[\text{sign}(\boldsymbol{B}+\tau\boldsymbol{Z})\odot\boldsymbol{\alpha}^2+\boldsymbol{\alpha}\odot\boldsymbol{Z}\big]^\top(\partial_1\eta(\boldsymbol{B}+\tau\boldsymbol{Z},\boldsymbol{\alpha}\tau))^2\Big\}$$

$$+\frac{1}{\tau}\mathbb{E}\Big\{\big[\text{sign}(\boldsymbol{B}+\tau\boldsymbol{Z})\odot\boldsymbol{\alpha}^2+\boldsymbol{\alpha}\odot\boldsymbol{Z}\big]^\top\big[(\eta(\boldsymbol{B}+\tau\boldsymbol{Z},\boldsymbol{\alpha}\tau)-\boldsymbol{B})\odot\partial_1^2\eta(\boldsymbol{B}+\tau\boldsymbol{Z},\boldsymbol{\alpha}\tau)\big]\Big\},$$

$$\text{(C.23)}$$

Now consider the first two terms on the right side of (C.23). We write $\mathsf{I}:=-\text{sign}(\boldsymbol{B}+\tau\boldsymbol{Z})$ . Then
using $0\le\partial_1\eta(\boldsymbol{B}+\tau\boldsymbol{Z},\boldsymbol{\alpha}\tau)\le 1$ where the inequalities hold elementwise,

$$\frac{1}{\tau^2}\mathbb{E}\Big\{\big[\boldsymbol{\alpha}\odot\mathsf{I}\odot(\eta(\boldsymbol{B}+\tau\boldsymbol{Z},\boldsymbol{\alpha}\tau)-\boldsymbol{B})\big]^\top\partial_1\eta(\boldsymbol{B}+\tau\boldsymbol{Z},\boldsymbol{\alpha}\tau)\Big\}-\frac{1}{\tau}\mathbb{E}\Big\{\big[\mathsf{I}\odot\boldsymbol{\alpha}^2+\boldsymbol{\alpha}\odot\boldsymbol{Z}\big]^\top(\partial_1\eta(\boldsymbol{B}+\tau\boldsymbol{Z},\boldsymbol{\alpha}\tau))^2\Big\}$$

$$\le\frac{1}{\tau^2}\mathbb{E}\Big\{\big[\boldsymbol{\alpha}\odot\mathsf{I}\odot(\eta(\boldsymbol{B}+\tau\boldsymbol{Z},\boldsymbol{\alpha}\tau)-\boldsymbol{B})\big]^\top\partial_1\eta(\boldsymbol{B}+\tau\boldsymbol{Z},\boldsymbol{\alpha}\tau)\Big\}-\frac{1}{\tau}\mathbb{E}\Big\{\big|\mathsf{I}\odot\boldsymbol{\alpha}^2+\boldsymbol{\alpha}\odot\boldsymbol{Z}\big|^\top\partial_1\eta(\boldsymbol{B}+\tau\boldsymbol{Z},\boldsymbol{\alpha}\tau)\Big\}$$

$$\le\frac{1}{\tau^2}\mathbb{E}\Big\{\big[\boldsymbol{\alpha}\odot\big(\mathsf{I}\odot(\eta(\boldsymbol{B}+\tau\boldsymbol{Z},\boldsymbol{\alpha}\tau)-\boldsymbol{B})-\tau|\mathsf{I}\odot\boldsymbol{\alpha}+\boldsymbol{Z}|\big)\big]^\top\partial_1\eta(\boldsymbol{B}+\tau\boldsymbol{Z},\boldsymbol{\alpha}\tau)\Big\}\le\boldsymbol{0},$$

$$\text{(C.24)}$$

where the final step of (C.24) follows since, elementwise, if $\eta(\boldsymbol{B}+\tau\boldsymbol{Z},\boldsymbol{\alpha}\tau)=\boldsymbol{0}$ then $\partial_1\eta(\boldsymbol{B}+\tau\boldsymbol{Z},\boldsymbol{\alpha}\tau)=\boldsymbol{0}$ and if $\eta(\boldsymbol{B}+\tau\boldsymbol{Z},\boldsymbol{\alpha}\tau)\ne\boldsymbol{0}$ then $\eta(\boldsymbol{B}+\tau\boldsymbol{Z},\boldsymbol{\alpha}\tau)=\boldsymbol{B}+\tau\boldsymbol{Z}-\boldsymbol{\alpha}\tau\text{sign}(\boldsymbol{B}+\tau\boldsymbol{Z})$
and therefore

$$\mathsf{I}\odot(\eta(\boldsymbol{B}+\tau\boldsymbol{Z},\boldsymbol{\alpha}\tau)-\boldsymbol{B})-\tau|\mathsf{I}\odot\boldsymbol{\alpha}+\boldsymbol{Z}|=\mathsf{I}\odot(\tau\boldsymbol{Z}-\boldsymbol{\alpha}\tau\text{sign}(\boldsymbol{B}+\tau\boldsymbol{Z}))-\tau|\mathsf{I}\odot\boldsymbol{\alpha}+\boldsymbol{Z}|$$

By considering the individual cases $\mathsf{I}=1\implies\boldsymbol{B}+\tau\boldsymbol{Z}<0$ and $\mathsf{I}=-1\implies\boldsymbol{B}+\tau\boldsymbol{Z}>0$, it is
not hard to see that the above is less than or equal to 0.

Now plugging (C.24) into (C.23), we have

$$\frac{\partial}{\partial \tau}\frac{1}{\tau}\mathbb{E}\Big\{\Big[\boldsymbol{\alpha}\odot\mathrm{sign}(\boldsymbol{B}+\tau\boldsymbol{Z})\odot(\eta(\boldsymbol{B}+\tau\boldsymbol{Z},\boldsymbol{\alpha}\tau)-\boldsymbol{B})\Big]^{\top}\partial_1\eta(\boldsymbol{B}+\tau\boldsymbol{Z},\boldsymbol{\alpha}\tau)\Big\}$$
$$\leq \frac{1}{\tau}\mathbb{E}\Big\{\Big[\mathrm{sign}(\boldsymbol{B}+\tau\boldsymbol{Z})\odot\boldsymbol{\alpha}^2+\boldsymbol{\alpha}\odot\boldsymbol{Z}\Big]^{\top}\Big[(\eta(\boldsymbol{B}+\tau\boldsymbol{Z},\boldsymbol{\alpha}\tau)-\boldsymbol{B})\odot\partial_1^2\eta(\boldsymbol{B}+\tau\boldsymbol{Z},\boldsymbol{\alpha}\tau)\Big]\Big\},$$
(C.25)

Next we note that $\partial_1^2\eta(\boldsymbol{B}+\tau\boldsymbol{Z},\boldsymbol{\alpha}\tau)$ is zero everywhere except at the 'kink' points $\boldsymbol{B}+\tau\boldsymbol{Z}=\boldsymbol{\alpha}\tau$ or $\boldsymbol{B}+\tau\boldsymbol{Z}=-\boldsymbol{\alpha}\tau$ and therefore when the inner product is elementwise non-zero we have that

$$\eta(\boldsymbol{B}+\tau\boldsymbol{Z},\boldsymbol{\alpha}\tau)-\boldsymbol{B}=\tau(\boldsymbol{\alpha}+\boldsymbol{Z})\odot\delta(\boldsymbol{B}+\tau\boldsymbol{Z}=-\boldsymbol{\alpha}\tau)+\tau(\boldsymbol{Z}-\boldsymbol{\alpha})\odot\delta(\boldsymbol{B}+\tau\boldsymbol{Z}=\boldsymbol{\alpha}\tau)$$
$$=-\boldsymbol{B}\odot[\delta(\boldsymbol{Z}=-\boldsymbol{B}/\tau-\boldsymbol{\alpha})+\delta(\boldsymbol{Z}=-\boldsymbol{B}/\tau+\boldsymbol{\alpha})].$$

This means

$$-\mathbb{E}\Big\{[\mathrm{sign}(\boldsymbol{B}+\tau\boldsymbol{Z})\odot\boldsymbol{\alpha}^2+\boldsymbol{\alpha}\odot\boldsymbol{Z}]^{\top}\Big[(\eta(\boldsymbol{B}+\tau\boldsymbol{Z},\boldsymbol{\alpha}\tau)-\boldsymbol{B})\odot\partial_1^2\eta(\boldsymbol{B}+\tau\boldsymbol{Z},\boldsymbol{\alpha}\tau)\Big]\Big\}$$
$$=-\mathbb{E}_{\boldsymbol{B}}\mathbb{E}_{\boldsymbol{Z}|\boldsymbol{B}}\Big\{[\mathrm{sign}(\boldsymbol{b}+\tau\boldsymbol{Z})\odot\boldsymbol{\alpha}^2+\boldsymbol{\alpha}\odot\boldsymbol{Z}]^{\top}\Big[(\eta(\boldsymbol{b}+\tau\boldsymbol{Z},\boldsymbol{\alpha}\tau)-\boldsymbol{b})\odot\partial_1^2\eta(\boldsymbol{b}+\tau\boldsymbol{Z},\boldsymbol{\alpha}\tau)\Big]\Big|\boldsymbol{B}=\boldsymbol{b}\Big\}$$
$$=-\mathbb{E}_{\boldsymbol{B}}\mathbb{E}_{\boldsymbol{Z}|\boldsymbol{B}}\Big\{[\boldsymbol{\alpha}^2+\boldsymbol{\alpha}\odot\boldsymbol{Z}]^{\top}\Big[\boldsymbol{B}\odot\delta(\boldsymbol{Z}=-\boldsymbol{B}/\tau-\boldsymbol{\alpha})\Big]+[-\boldsymbol{\alpha}^2+\boldsymbol{\alpha}\odot\boldsymbol{Z}]^{\top}\Big[\boldsymbol{B}\odot\delta(\boldsymbol{Z}=-\boldsymbol{B}/\tau+\boldsymbol{\alpha})\Big]\Big|\boldsymbol{B}=\boldsymbol{b}\Big\}$$
$$=\frac{1}{\tau}\mathbb{E}_{\boldsymbol{B}}\Big\{[\boldsymbol{\alpha}\odot(\boldsymbol{B}/\tau))]^{\top}[\boldsymbol{B}\odot\phi(\boldsymbol{B}/\tau+\boldsymbol{\alpha})]+[\boldsymbol{\alpha}\odot(\boldsymbol{B}/\tau)]^{\top}[\boldsymbol{B}\odot\phi(-\boldsymbol{B}/\tau+\boldsymbol{\alpha})]\Big\}$$
$$=\frac{1}{\tau^2}\mathbb{E}_{\boldsymbol{B}}\Big\{[\boldsymbol{\alpha}\odot\boldsymbol{B}^2]^{\top}[\phi(\boldsymbol{B}/\tau+\boldsymbol{\alpha})+\phi(-\boldsymbol{B}/\tau+\boldsymbol{\alpha})]\Big\}.$$
(C.26)

Now we plug (C.19),(C.20) and (C.26) back into (C.15) in order to show that $\frac{\partial}{\partial\tau}\big[\frac{\partial\mathsf{F}}{\partial\tau^2}(\tau^2,\boldsymbol{\alpha}\tau)\big]\leq 0$.

$$(\delta p)\times\frac{\partial}{\partial\tau}\Big[\frac{\partial\mathsf{F}}{\partial\tau^2}(\tau^2,\boldsymbol{\alpha}\tau)\Big]\leq-\frac{1}{\tau^4}\mathbb{E}_{\boldsymbol{B}}\Big\{(\boldsymbol{B}^3)^{\top}[\phi(-\boldsymbol{B}/\tau+\boldsymbol{\alpha})-\phi(\boldsymbol{B}/\tau+\boldsymbol{\alpha})]\Big\}.$$
(C.27)

Now we justify that the above is non-positive by showing that the elementwise term inside the expectation is non-positive. First assume $B_i\geq 0$, then $\alpha_i-B_i/\tau\leq\alpha_i+B_i/\tau$ and $\phi(\alpha_i-B_i/\tau)\geq\phi(\alpha_i+B_i/\tau)$. The other case $B_i\leq 0$ follows similarly.

Therefore, (C.27), implies

$$\frac{\partial}{\partial\tau}\Big[\frac{\partial\mathsf{F}}{\partial\tau^2}(\tau^2,\boldsymbol{\alpha}\tau)\Big]\leq 0$$

Recall in (C.13), we suppressed $|I_i|$ to 1 and consequently set $\sum_{j\in I_i}\alpha_j=\alpha_i$. Now we validate that we did not lose generality: multiplying $|I_i|$ to (C.27) does not change the negativity as shown, and changing $\alpha_i\to\sum_{j\in I_i}\alpha_j$ makes no difference to the conclusion as both terms are positive.

Now we have shown that $\mathsf{F}(\tau^2,\boldsymbol{\alpha}\tau)$ defined in (2.8), is concave with respect to $\tau^2$. Next we consider taking $\tau^2\to\infty$. Define $f(\boldsymbol{\alpha}):=\delta\lim_{\tau\to\infty}\frac{d\mathsf{F}}{d\tau^2}(\tau^2,\boldsymbol{\alpha}\tau)$. By a very similar argument to the proof of concavity, we can see $f'(\boldsymbol{\alpha})<0$: letting $\tau\to\infty$ is equivalent to setting $\boldsymbol{B}=\mathbf{0}$ in (C.10). Now we define $\boldsymbol{D}$ elementwise as $[\boldsymbol{D}(\boldsymbol{v})]_i:=\#\{j:|v_j|=|v_i|\}$ if $v_i\neq 0$ and $\infty$ otherwise, define $\boldsymbol{\eta}:=\eta(\boldsymbol{Z},\boldsymbol{\alpha})$ and rewrite $\partial_1\eta(\tau\boldsymbol{Z},\boldsymbol{\alpha}\tau)=\partial_1\eta(\boldsymbol{Z},\boldsymbol{\alpha})=\frac{1}{\boldsymbol{D}(\boldsymbol{\eta})}$. From (C.10),

$$f(\boldsymbol{\alpha})=\frac{1}{p}\sum_{i=1}^{p}\mathbb{E}\Big\{[\boldsymbol{D}(\eta(\boldsymbol{Z},\boldsymbol{\alpha}))]_i([\partial_1\eta(\boldsymbol{Z},\boldsymbol{\alpha})]_i)^2-[\partial_1\eta(\boldsymbol{Z},\boldsymbol{\alpha})]_i\,|[\eta(\boldsymbol{Z},\boldsymbol{\alpha})]_i|\sum_{j:|[\eta(\boldsymbol{Z},\boldsymbol{\alpha})]_i|=|[\eta(\boldsymbol{Z},\boldsymbol{\alpha})]_i|}\alpha_j\Big\}$$
$$=\frac{1}{p}\sum_{i=1}^{p}\mathbb{E}\left\{\left(1-|[\eta(\boldsymbol{Z},\boldsymbol{\alpha})]_i|\sum_{j:|[\eta(\boldsymbol{Z},\boldsymbol{\alpha})]_i|=|[\eta(\boldsymbol{Z},\boldsymbol{\alpha})]_i|}\alpha_j\right)/[\boldsymbol{D}(\eta(\boldsymbol{Z},\boldsymbol{\alpha}))]_i\right\}$$
(C.28)

This simplification is both explicit and much more efficient in computation because only $\boldsymbol{\eta}$ need to be memorized.

Now considering the above we let $\alpha \to \infty$ and we note that in this case $\eta(\boldsymbol{Z}, \boldsymbol{\alpha}) = \partial_1 \eta(\boldsymbol{Z}, \boldsymbol{\alpha}) = \boldsymbol{0}$ since $\boldsymbol{Z} < \boldsymbol{\alpha}$ almost surely as $\boldsymbol{\alpha} \to \infty$. Therefore $\lim_{\boldsymbol{\alpha} \to \infty} f(\boldsymbol{\alpha}) = \boldsymbol{0}$. Combined with the fact that $f'(\boldsymbol{\alpha}) < 0$, it follows that $f(\boldsymbol{\alpha}) > 0$ for all $\boldsymbol{\alpha}$.

Since $\mathsf{F}$ is concave and strictly increasing for $\tau^2$ large enough, it is increasing everywhere. Another fact is that by (C.28), we know that

$$f(\boldsymbol{0}) = \frac{1}{p}\mathbb{E}\|\partial_1 \eta(\tau \boldsymbol{Z}, \boldsymbol{0})\|^2 = \frac{1}{p}\mathbb{E}\|\partial_1(\tau \boldsymbol{Z})\|^2 = \frac{1}{p}\mathbb{E}\|\boldsymbol{Z}\|^2 = 1.$$

Then considering the function $\mathsf{F}$ again, where

$$\mathsf{F}(\tau^2, \boldsymbol{\alpha}\tau) := \sigma^2 + \frac{1}{\delta p}\mathbb{E}\|\mathrm{prox}_{J_{\alpha\tau}}(\boldsymbol{B} + \tau \boldsymbol{Z}) - \boldsymbol{B}\|^2,$$

we have $\mathsf{F}(0,0) = \sigma^2 \geq 0$, and $\frac{d\mathsf{F}}{d\tau^2}(0,0) = \frac{1}{\delta} \geq 1$. Comparing to $\tau^2 = 0, \frac{d\tau^2}{d\tau^2} = 1$ at $\tau = 0$, together with $f(\infty) = 0$, the fixed point equation admits at least one solution for $\boldsymbol{\alpha}$ satisfying $f(\boldsymbol{\alpha}) < \delta$. Notice by monotonicity of $f$, we can define $A_{\min} := \{\boldsymbol{\alpha} : f(\boldsymbol{\alpha}) = \delta\}$ and requires $\boldsymbol{\alpha}$ being larger than at least one element in $A_{\min}$. Here a vector $\boldsymbol{v}$ is larger than another vector $\boldsymbol{u}$ when $\forall i, v_i \geq u_i$ and $v_j > u_j$ for some $j$. The concavity of $\mathsf{F}$ guarantees that the solution is unique and that the sequence of iterates $\tau_t$ converge to $\tau$. Finally, at fixed point, $\frac{d\mathsf{F}}{d\tau^2}(\tau, \alpha\tau) \leq \frac{d\tau^2}{d\tau^2} = 1$. $\qquad \square$

# D   Verifying Properties (P1) and (P2)

In this appendix we demonstrate that the properties **(P1)** and **(P2)** given in Section B and relating to the denoiser $\eta_p^t(\cdot)$ defined in (B.1) are true.

*Verifying Properties **(P1)** and **(P2)**.* Property **(P1)** follows from the fact that $\eta_p^t(\cdot) = \mathrm{prox}_{J_{\alpha\tau_t}}(\cdot)$, as it is easy to show that proximal operators are Lipschitz continuous with Lipschitz constant one. Namely

$$\|\eta_p^t(\boldsymbol{v}_1) - \eta_p^t(\boldsymbol{v}_2)\| = \|\mathrm{prox}_{J_{\alpha\tau_t}}(\boldsymbol{v}_1) - \mathrm{prox}_{J_{\alpha\tau_t}}(\boldsymbol{v}_2)\| \leq \|\boldsymbol{v}_1 - \boldsymbol{v}_2\|.$$

Next we show that property **(P2)** is true. We restate property **(P2)** for convenience: for any $s, t$ with $(\boldsymbol{Z}, \boldsymbol{Z}')$ a pair of length-$p$ vectors such that $(Z_i, Z_i')$ are i.id. $\sim \mathcal{N}(0, \boldsymbol{\Sigma})$ for $i \in [p]$ where $\boldsymbol{\Sigma}$ is any $2 \times 2$ covariance matrix, the following limits exist and are finite.

$$\mathop{\mathrm{plim}}_{p \to \infty} \frac{1}{p}\|\boldsymbol{\beta}\|, \tag{D.1}$$

$$\mathop{\mathrm{plim}}_{p \to \infty} \frac{1}{p}\mathbb{E}_{\boldsymbol{Z}}[\boldsymbol{\beta}^\top \eta_p^t(\boldsymbol{\beta} + \boldsymbol{Z})], \tag{D.2}$$

$$\mathop{\mathrm{plim}}_{p \to \infty} \frac{1}{p}\mathbb{E}_{\boldsymbol{Z}, \boldsymbol{Z}'}[\eta_p^s(\boldsymbol{\beta} + \boldsymbol{Z}')^\top \eta_p^t(\boldsymbol{\beta} + \boldsymbol{Z})]. \tag{D.3}$$

We first note that the limit in (D.1) exists by Assumption **(A2)** and the strong law of large numbers. We focus on the other two limits. These results follow by [21, Proposition 1] given in Lemma E.1 and the following lemma, which is a classic result in probability theory.

**Lemma D.1** (Doob's $L^1$ maximal inequality, [19] Chapter VII, Theorem 3.4). *Let $X_1, X_2, \ldots, X_p$ be a sequence of nonnegative i.i.d. random variables such that $\mathbb{E}[X_1 \max\{0, \log(X_1)\}] < \infty$. Then,*

$$\mathbb{E}\left[\sup_{p \geq 1}\left\{\frac{X_1 + X_2 + \cdots + X_p}{p}\right\}\right] \leq \frac{e}{e-1}(1 + \mathbb{E}[X_1 \max\{0, \log(X_1)\}]).$$

*Proof.* Let $M_p = \frac{1}{p}(X_1 + X_2 + \cdots + X_p)$. Then the sequence $\{M_p\}$ is a submartingale and hence by Doob's maximal inequality,

$$\mathbb{E}\left[\sup_{p' \geq p \geq 1} M_p\right] \leq \frac{e}{e-1}(1 + \mathbb{E}[M_{p'} \max\{0, \log(M_{p'})\}]).$$

Note the mapping $x \mapsto x \max\{0, \log x\}$ is convex and hence $\mathbb{E}[M_{p'} \max\{0, \log(M_{p'})\}]) \leq \mathbb{E}[X_1 \max\{0, \log(X_1)\}]$. The result follows by Fatou's lemma and by noting that $\sup_{p' \geq p \geq 1} M_p \uparrow \sup_{p \geq 1} M_p$ as $p' \to \infty$. $\qquad\square$

We first consider the limit in (D.2). Note that by Cauchy-Schwarz, (E.1) implies that

$$\frac{1}{p} \left| \boldsymbol{\beta}^\top \eta_p^t(\boldsymbol{\beta} + \boldsymbol{Z}) - \boldsymbol{\beta}^\top h^t(\boldsymbol{\beta} + \boldsymbol{Z}) \right| \to 0, \quad \text{as} \quad p \to \infty, \tag{D.4}$$

This follows because

$$\frac{1}{p} \left| \boldsymbol{\beta}^\top \eta_p^t(\boldsymbol{\beta} + \boldsymbol{Z}) - \boldsymbol{\beta}^\top h^t(\boldsymbol{\beta} + \boldsymbol{Z}) \right| \leq \frac{\|\boldsymbol{\beta}\|}{\sqrt{p}} \times \frac{\|\eta_p^t(\boldsymbol{\beta} + \boldsymbol{Z}) - h^t(\boldsymbol{\beta} + \boldsymbol{Z})\|}{\sqrt{p}}.$$

Then the right side of the above goes to 0 with growing $p$ because $\|\boldsymbol{\beta}\| / \sqrt{p}$ limits to a constant as justified above (this is the limit in (D.1)), and the other term limits to 0 by (E.1).

Next, we show that $\mathbb{E}_{\boldsymbol{Z}}\{\sup_{p \geq 1} \frac{1}{p} \left| \boldsymbol{\beta}^\top \eta_p^t(\boldsymbol{\beta} + \boldsymbol{Z}) - \boldsymbol{\beta}^\top h^t(\boldsymbol{\beta} + \boldsymbol{Z}) \right|\}$ is finite. Since both $\eta_p^t$ and $h^t$ are Lipschitz(1), by Cauchy-Schwarz inequality, the desired expectation is finite almost surely if both

$$\mathbb{E}\left[\sup_{p \geq 1}\left\{\frac{\|\boldsymbol{Z}(p)\|^2}{p}\right\}\right] < \infty \quad \text{and} \quad \mathbb{E}\left[\sup_{p \geq 1}\left\{\frac{\|\boldsymbol{B}(p)\|^2}{p}\right\}\right] < \infty.$$

But Lemma D.1 immediately implies the above since $\mathbb{E}[B^2 \max\{0, \log B\}] < \infty$ by assumption **(A2)**.

Now by dominated convergence we have that

$$\mathbb{E}_{\boldsymbol{Z}}\left\{\lim_p \frac{1}{p} \left| \boldsymbol{\beta}^\top \eta_p^t(\boldsymbol{\beta} + \boldsymbol{Z}) - \boldsymbol{\beta}^\top h^t(\boldsymbol{\beta} + \boldsymbol{Z}) \right|\right\} = \lim_p \frac{1}{p} \mathbb{E}_{\boldsymbol{Z}} \left| \boldsymbol{\beta}^\top \eta_p^t(\boldsymbol{\beta} + \boldsymbol{Z}) - \boldsymbol{\beta}^\top h^t(\boldsymbol{\beta} + \boldsymbol{Z}) \right|$$

$$\geq \lim_p \frac{1}{p} \left| \mathbb{E}_{\boldsymbol{Z}}\{\boldsymbol{\beta}^\top \eta_p^t(\boldsymbol{\beta} + \boldsymbol{Z})\} - \mathbb{E}_{\boldsymbol{Z}}\{\boldsymbol{\beta}^\top h^t(\boldsymbol{\beta} + \boldsymbol{Z})\} \right|.$$

Then the above implies that

$$\frac{1}{p} \left| \mathbb{E}_{\boldsymbol{Z}}\{\boldsymbol{\beta}^\top \eta_p^t(\boldsymbol{\beta} + \boldsymbol{Z})\} - \mathbb{E}_{\boldsymbol{Z}}\{\boldsymbol{\beta}^\top h^t(\boldsymbol{\beta} + \boldsymbol{Z})\} \right| \to 0, \quad \text{as} \quad p \to \infty.$$

Therefore,

$$\plim_{p \to \infty} \frac{1}{p} \mathbb{E}_{\boldsymbol{Z}}[\boldsymbol{\beta}^\top \eta_p^t(\boldsymbol{\beta} + \boldsymbol{Z})] = \plim_{p \to \infty} \frac{1}{p} \left( \beta_{0,1} \mathbb{E}_Z\{h^t(\beta_{0,1} + Z_1)\} + \cdots + \beta_{0,p} \mathbb{E}_Z\{h^t(\beta_{0,p} + Z_p)\} \right)$$

$$= \mathbb{E}[Bh^t(B + Z)],$$

where $B, Z$ are univariate. By Cauchy Schwarz inequality, $\mathbb{E}[Bh^t(B + Z)] < \infty$ if $\mathbb{E}[B^2] < \infty$ and $\mathbb{E}[h^t(B + Z)^2] < \infty$. Since $\mathbb{E}[B^2] = \sigma_{\boldsymbol{\beta}}^2 < \infty$ is given by our assumption, it suffices to show $\mathbb{E}[h^t(B + Z)^2] < \infty$. But this follows from the fact that $h^t(\cdot)$ is Lipschitz(1) and therefore $\mathbb{E}[h^t(B + Z)^2] < \mathbb{E}[(B + Z)^2] \leq \mathbb{E}[B^2] + \mathbb{E}[Z^2] = \sigma_{\boldsymbol{\beta}}^2 + \Sigma_{11} < \infty$.

Finally consider (D.3). Similarly to the work in studying the limit in (D.2), we can show

$$\plim_{p \to \infty} \frac{1}{p} \mathbb{E}[\eta_p^s(\boldsymbol{\beta} + \boldsymbol{Z}')^\top \eta_p^t(\boldsymbol{\beta} + \boldsymbol{Z})] = \mathbb{E}[h^s(B + Z')h^t(B + Z)], \tag{D.5}$$

where $\mathbb{E}[h^s(B + Z')h^t(B + Z)] < \infty$ by Cauchy-Schwarz and the fact that $h^s(\cdot)$ and $h^t(\cdot)$ are Lipschitz(1). Namely, this gives the bound

$$\left( \mathbb{E}[h^s(B + Z')h^t(B + Z)] \right)^2 \leq \mathbb{E}[(h^s(B + Z'))^2] \, \mathbb{E}[(h^t(B + Z))^2] \leq \mathbb{E}[(B + Z')^2] \, \mathbb{E}[(B + Z)^2]$$

$$= (\mathbb{E}[B^2] + \mathbb{E}[Z'^2])(\mathbb{E}[B^2] + \mathbb{E}[Z^2]) = (\sigma_{\boldsymbol{\beta}}^2 + \Sigma_{22})(\sigma_{\boldsymbol{\beta}}^2 + \Sigma_{11}) < \infty.$$

In order to prove (D.5), we will show that

$$\frac{1}{p} \left| \eta_p^s(\boldsymbol{\beta} + \boldsymbol{Z}')^\top \eta_p^t(\boldsymbol{\beta} + \boldsymbol{Z}) - h^s(\boldsymbol{\beta} + \boldsymbol{Z}')^\top h^t(\boldsymbol{\beta} + \boldsymbol{Z}) \right| \to 0, \quad \text{as} \quad p \to \infty, \tag{D.6}$$

Then (D.5) follows from (D.6) using a dominated convergence argument, i.e. since both $\eta_p^t(\cdot)$ and $h^t(\cdot)$ are Lipschitz(1), using the Cauchy-Schwarz inequality, if both

$$\mathbb{E}\left[\sup_{p\geq 1}\left\{\frac{\|\boldsymbol{Z}(p)\|^2}{p}\right\}\right] < \infty \quad \text{and} \quad \mathbb{E}\left[\sup_{p\geq 1}\left\{\frac{\|\boldsymbol{B}(p)\|^2}{p}\right\}\right] < \infty. \tag{D.7}$$

then

$$\mathop{\mathbb{E}}_{\boldsymbol{Z},\boldsymbol{Z}'}\left\{\sup_{p\geq 1}\frac{1}{p}\left|\eta_p^s(\boldsymbol{\beta}+\boldsymbol{Z}')^\top\eta_p^t(\boldsymbol{\beta}+\boldsymbol{Z}) - h^s(\boldsymbol{\beta}+\boldsymbol{Z}')^\top h^t(\boldsymbol{\beta}+\boldsymbol{Z})\right|\right\} < \infty. \tag{D.8}$$

Note that as before, (D.7) is true by Lemma D.1 and assumption (**A2**), and so we have the result in (D.8). Now by dominated convergence, we have that

$$\mathop{\mathbb{E}}_{\boldsymbol{Z},\boldsymbol{Z}'}\left\{\lim_p\frac{1}{p}\left|\eta_p^s(\boldsymbol{\beta}+\boldsymbol{Z}')^\top\eta_p^t(\boldsymbol{\beta}+\boldsymbol{Z}) - h^s(\boldsymbol{\beta}+\boldsymbol{Z}')^\top h^t(\boldsymbol{\beta}+\boldsymbol{Z})\right|\right\}$$

$$= \lim_p\frac{1}{p}\mathop{\mathbb{E}}_{\boldsymbol{Z},\boldsymbol{Z}'}\left|\eta_p^s(\boldsymbol{\beta}+\boldsymbol{Z}')^\top\eta_p^t(\boldsymbol{\beta}+\boldsymbol{Z}) - h^s(\boldsymbol{\beta}+\boldsymbol{Z}')^\top h^t(\boldsymbol{\beta}+\boldsymbol{Z})\right|$$

$$\geq \lim_p\frac{1}{p}\left|\mathop{\mathbb{E}}_{\boldsymbol{Z},\boldsymbol{Z}'}\{\eta_p^s(\boldsymbol{\beta}+\boldsymbol{Z}')^\top\eta_p^t(\boldsymbol{\beta}+\boldsymbol{Z})\} - \mathop{\mathbb{E}}_{\boldsymbol{Z},\boldsymbol{Z}'}\{h^s(\boldsymbol{\beta}+\boldsymbol{Z}')^\top h^t(\boldsymbol{\beta}+\boldsymbol{Z})\}\right|.$$

Then the above implies, along with the result in (D.6), that

$$\frac{1}{p}\left|\mathop{\mathbb{E}}_{\boldsymbol{Z},\boldsymbol{Z}'}\{\eta_p^s(\boldsymbol{\beta}+\boldsymbol{Z}')^\top\eta_p^t(\boldsymbol{\beta}+\boldsymbol{Z})\} - \mathop{\mathbb{E}}_{\boldsymbol{Z},\boldsymbol{Z}'}\{h^s(\boldsymbol{\beta}+\boldsymbol{Z}')^\top h^t(\boldsymbol{\beta}+\boldsymbol{Z})\}\right| \to 0, \quad \text{as} \quad p \to \infty.$$

But now, using the above, we have the desired result (D.5). Namely,

$$\mathop{\text{plim}}_{p\to\infty}\frac{1}{p}\mathop{\mathbb{E}}_{\boldsymbol{Z},\boldsymbol{Z}'}\{\eta_p^s(\boldsymbol{\beta}+\boldsymbol{Z}')^\top\eta_p^t(\boldsymbol{\beta}+\boldsymbol{Z})\} = \mathop{\text{plim}}_{p\to\infty}\frac{1}{p}\sum_{i=1}^p\mathop{\mathbb{E}}_{\boldsymbol{Z},\boldsymbol{Z}'}\{h^s(\beta_i+Z_i')h^t(\beta_i+Z_i)\}$$

$$= \mathbb{E}[h^s(B+Z')h^t(B+Z)],$$

where $B, Z'$, and $Z$ are univariate. We have now shown (D.5).

Now we want to prove (D.6). First note

$$\left|\eta_p^s(\boldsymbol{\beta}+\boldsymbol{Z}')^\top\eta_p^t(\boldsymbol{\beta}+\boldsymbol{Z}) - h^s(\boldsymbol{\beta}+\boldsymbol{Z}')^\top h^t(\boldsymbol{\beta}+\boldsymbol{Z})\right|$$

$$\leq \left|\eta_p^s(\boldsymbol{\beta}+\boldsymbol{Z}')^\top\eta_p^t(\boldsymbol{\beta}+\boldsymbol{Z}) - h^s(\boldsymbol{\beta}+\boldsymbol{Z}')^\top\eta_p^t(\boldsymbol{\beta}+\boldsymbol{Z})\right| + \left|h^s(\boldsymbol{\beta}+\boldsymbol{Z}')^\top\eta_p^t(\boldsymbol{\beta}+\boldsymbol{Z}) - h^s(\boldsymbol{\beta}+\boldsymbol{Z}')^\top h^t(\boldsymbol{\beta}+\boldsymbol{Z})\right|. \tag{D.9}$$

Now we upper bound both of the terms on the right side of (D.9) using Cauchy-Schwarz. Consider the second term on the right side of (D.9). Then,

$$\left|h^s(\boldsymbol{\beta}+\boldsymbol{Z}')^\top\eta_p^t(\boldsymbol{\beta}+\boldsymbol{Z}) - h^s(\boldsymbol{\beta}+\boldsymbol{Z}')^\top h^t(\boldsymbol{\beta}+\boldsymbol{Z})\right| \leq \|h^s(\boldsymbol{\beta}+\boldsymbol{Z}')\|\|\eta_p^t(\boldsymbol{\beta}+\boldsymbol{Z}) - h^t(\boldsymbol{\beta}+\boldsymbol{Z})\|. \tag{D.10}$$

Now consider the first term on the right side of (D.9). Again, by Cauchy-Scwarz,

$$\left|\eta_p^s(\boldsymbol{\beta}+\boldsymbol{Z}')^\top\eta_p^t(\boldsymbol{\beta}+\boldsymbol{Z}) - h^s(\boldsymbol{\beta}+\boldsymbol{Z}')^\top\eta_p^t(\boldsymbol{\beta}+\boldsymbol{Z})\right| = \left|\left[\eta_p^s(\boldsymbol{\beta}+\boldsymbol{Z}') - h^s(\boldsymbol{\beta}+\boldsymbol{Z}')\right]^\top\eta_p^t(\boldsymbol{\beta}+\boldsymbol{Z})\right|$$

$$= \left|\left[\eta_p^s(\boldsymbol{\beta}+\boldsymbol{Z}') - h^s(\boldsymbol{\beta}+\boldsymbol{Z}')\right]^\top\left[\eta_p^t(\boldsymbol{\beta}+\boldsymbol{Z}) - h^t(\boldsymbol{\beta}+\boldsymbol{Z}) + h^t(\boldsymbol{\beta}+\boldsymbol{Z})\right]\right|$$

$$\leq \left|\left[\eta_p^s(\boldsymbol{\beta}+\boldsymbol{Z}') - h^s(\boldsymbol{\beta}+\boldsymbol{Z}')\right]^\top\left[\eta_p^t(\boldsymbol{\beta}+\boldsymbol{Z}) - h^t(\boldsymbol{\beta}+\boldsymbol{Z})\right]\right| + \left|\left[\eta_p^s(\boldsymbol{\beta}+\boldsymbol{Z}') - h^s(\boldsymbol{\beta}+\boldsymbol{Z}')\right]^\top h^t(\boldsymbol{\beta}+\boldsymbol{Z})\right|$$

$$\leq \|\eta_p^s(\boldsymbol{\beta}+\boldsymbol{Z}') - h^s(\boldsymbol{\beta}+\boldsymbol{Z}')\|\|\eta_p^t(\boldsymbol{\beta}+\boldsymbol{Z}) - h^t(\boldsymbol{\beta}+\boldsymbol{Z})\| + \|h^t(\boldsymbol{\beta}+\boldsymbol{Z})\|\|\eta_p^s(\boldsymbol{\beta}+\boldsymbol{Z}') - h^s(\boldsymbol{\beta}+\boldsymbol{Z}')\|. \tag{D.11}$$

Plugging (D.10) and (D.11) into (D.9) we find

$$\lim_p\frac{1}{p}\left|\eta_p^s(\boldsymbol{\beta}+\boldsymbol{Z}')^\top\eta_p^t(\boldsymbol{\beta}+\boldsymbol{Z}) - h^s(\boldsymbol{\beta}+\boldsymbol{Z}')^\top h^t(\boldsymbol{\beta}+\boldsymbol{Z})\right|$$

$$\leq \lim_p\frac{\|\eta_p^s(\boldsymbol{\beta}+\boldsymbol{Z}') - h^s(\boldsymbol{\beta}+\boldsymbol{Z}')\|}{\sqrt{p}}\frac{\|\eta_p^t(\boldsymbol{\beta}+\boldsymbol{Z}) - h^t(\boldsymbol{\beta}+\boldsymbol{Z})\|}{\sqrt{p}} + \lim_p\frac{\|h^t(\boldsymbol{\beta}+\boldsymbol{Z})\|}{\sqrt{p}}\frac{\|\eta_p^s(\boldsymbol{\beta}+\boldsymbol{Z}') - h^s(\boldsymbol{\beta}+\boldsymbol{Z}')\|}{\sqrt{p}}$$

$$+ \lim_p\frac{\|h^s(\boldsymbol{\beta}+\boldsymbol{Z}')\|}{\sqrt{p}}\frac{\|\eta_p^t(\boldsymbol{\beta}+\boldsymbol{Z}) - h^t(\boldsymbol{\beta}+\boldsymbol{Z})\|}{\sqrt{p}}.$$

Now, (D.6) follows since the right side of the above goes to 0 as $p$ grows. This follows since, by (E.1), as $p \to \infty$,

$$\frac{\|\eta_p^s(\boldsymbol{\beta} + \boldsymbol{Z}') - h^s(\boldsymbol{\beta} + \boldsymbol{Z}')\|}{\sqrt{p}} \to 0 \quad \text{and} \quad \frac{\|\eta_p^t(\boldsymbol{\beta} + \boldsymbol{Z}) - h^t(\boldsymbol{\beta} + \boldsymbol{Z})\|}{\sqrt{p}} \to 0.$$

Moreover, since $h^s(\cdot)$ and $h^t(\cdot)$ are separable, by the Law of Large Numbers,

$$\lim_p \frac{\|h^s(\boldsymbol{\beta} + \boldsymbol{Z}')\|^2}{p} = \lim_p \frac{1}{p} \sum_{i=1}^p [h^s(\beta_i + Z_i')]^2 = \mathbb{E}[(h^s(B + Z'))^2] < \infty,$$

$$\lim_p \frac{\|h^t(\boldsymbol{\beta} + \boldsymbol{Z})\|^2}{p} = \lim_p \frac{1}{p} \sum_{i=1}^p [h^t(\beta_i + Z_i)]^2 = \mathbb{E}[(h^t(B + Z))^2] < \infty,$$

where the inequalities follow since $\mathbb{E}[(h^s(B + Z'))^2] \le \mathbb{E}[(B + Z')^2] \le \sigma_{\boldsymbol{\beta}}^2 + \Sigma_{22} < \infty$ and $\mathbb{E}[(h^t(B + Z))^2] \le \mathbb{E}[(B + Z)^2] \le \sigma_{\boldsymbol{\beta}}^2 + \Sigma_{11} < \infty$ We have now shown that property **(P2)** is true.

□

## E  Proof of Fact 2.4

*Proof.* The fact follows from the asymptotic separability of the proximal operator [21, Proposition 1] and the dominated convergence theorem [30] allowing for interchange of limit and expectation. We sketch this argument now, but first restate [21, Proposition 1], which says that $\text{prox}_{J_{\alpha \tau_t}}(\cdot)$ becomes asymptotically separable as $p \to \infty$, for convenience.

**Lemma E.1** (Proposition 1, [21]). *For an input sequence $\{\boldsymbol{v}(p)\}$, and a sequence of thresholds $\{\boldsymbol{\lambda}(p)\}$, both having empirical distributions that weakly converge to a distributions $V$ and $\Lambda$, respectively, then there exists a limiting scalar function $h$ (determined by $V$ and $\Lambda$) such that as $p \to \infty$,*

$$\frac{1}{p} \|\text{prox}_{J_{\boldsymbol{\lambda}(p)}}(\boldsymbol{v}(p)) - h(\boldsymbol{v}(p); V, \Lambda)\|^2 \to 0, \tag{E.1}$$

*where $h$ applies $h(\cdot; V, \Lambda)$ coordinate-wise to $\boldsymbol{v}(p)$ (hence it is separable) and $h$ is Lipschitz(1).*

Now we sketch the proof of the existence of the limit in (2.4) (and the result for the limit in (2.10) follows similarly). By Lemma E.1, the weak convergence of $\boldsymbol{\alpha}(p)$ to $A$, and the Weak Law of Large Numbers, one can argue that

$$\lim_p \frac{1}{\delta p} \|\text{prox}_{J_{\boldsymbol{\alpha}(p)\tau_*}}(\boldsymbol{B} + \tau_* \boldsymbol{Z}) - \boldsymbol{B}\|^2 = \frac{1}{\delta} \mathbb{E}(h(B + \tau_* Z) - B)^2, \tag{E.2}$$

where $h(\cdot) := h(\cdot; B + \tau_* Z, A\tau_*)$ is the unspecified, separable function of Lemma E.1. This is consistent with [Lemma 29, [21]]. So then the limit in (2.4) exists if $\frac{1}{\delta} \mathbb{E}(h(B + \tau_* Z) - B)^2 < \infty$ which is true:

$$\mathbb{E}(h(B + \tau_* Z) - B)^2 \le 2\mathbb{E}(h(B + \tau_* Z)^2 + B^2) \le 2\mathbb{E}((B + \tau_* Z)^2 + B^2)$$
$$\le 2\mathbb{E}(2B^2 + 2\tau_*^2 Z^2 + B^2) = 6\mathbb{E}(B^2) + 4\tau_*^2 < \infty.$$

Here the first and third inequalities follow from $(x - y)^2 \le 2(x^2 + y^2)$ and the second inequality follows from $h$ being Lipschitz(1): $|h(x)| = |h(x) - h(0)| \le |x - 0| = |x|$.

□