[Reviews · NeurIPS 2019]

Reviewer 1



The main contribution of this work is to show that the approximate message passing (AMP) algorithm, which has originally been developed in the context of losses with separable penalties, can be adapted for a specific non-separable penalty. It moreover focus on proving that the resulting algorithm successfully converges to the correct solution. It is important to mention that the focus here is on providing an AMP-based solution to the problem, as similar results for this penalty had already been obtained using different techniques in [20]. The mathematical analysis performed seems sound, and while the framework used has already been introduced in [7], new techniques were employed in order to deal with the possibly non-strongly convex loss. While AMP algorithms have gained prominence in the last decade, there remains much work to be done before they become practical, mostly on providing convergence guarantees for arbitrarily distributed data. Adapting AMP to specific losses is important, but does not necessarily translate into a big impact for the machine learning community. As such, I believe this work belongs to more specialized venues rather than at NeurIPS. Minor points follow: - the notation $|b|_{(i)}$ is not clear; order statistics should be defined to the unfamiliar reader - typo on caption for Fig/Table 1: "hasi.i.d." - on page 2: emphasize experiments are on Gaussian i.i.d. data # Post-rebuttal addendum I believe the authors have put a great deal of effort on the rebuttal, answering most of the points raised in the reviews. Moreover, I am impressed they were able to come up with a SLOPE variant of VAMP in such a short notice. This new algorithm, together with remaining results presented, addresses my concerns regarding the usefulness of the proposed approach. I have thus decided to increase my score to 7, in spite of my belief that the paper is perhaps a bit too specific. I recommend the authors update the manuscript with the information contained in the rebuttal, in particular regarding the possibility of having alternative message-passing algorithms that are more stable in practice.

Reviewer 2



This is a solid paper in the line of work that provides sharp results for linear estimation problems with random iid matrices/designs. It extends the approximate message passing algorithm and the proof of its state evolution to the non-separable penalty called SLOPE. At the same time as the author note, AMP has already been extended to non-separable penalties, e.g. [7], and the performance of the SLOPE penalty was already analysed using other methods recently (ref. [20].). This together with the results on [12] that show limitations of SLOPE common to all convex penalties, limits somewhat the originality and significance of the work. Suggestions: ** It could be useful to the readers if the authors discuss in related work the main results of ref. [12] to highlight the gap that is obtained between optimal estimators and any convex penalty. ** The discussion of AMP applied to non-separable penalties could be extended, including for instance Manoel, Krzakala, Varoquaux, Thirion, Zdeborova, "Approximate message-passing for convex optimization with non-separable penalties" --------- Post-feedback: I have read the feedback and the other reviews. I maintain my score. The paper presents theoretical contributions of interest towards analysis of AMP with non-separable convex penalties. But I do not consider the theoretical contribution groundbreaking and I consider the algorithm limited in applicability because of the convexity of the penalty and the assumptions on the matrix.

Reviewer 3



The submission is original and very clearly written. The contribution is significant if one is a theoretician and okay with large iid Gaussian designs and iid ground truth "beta" with known distribution "B". Indeed, the work is mathematically deep compared to typical NeurIPS papers. The numerical simulations show that AMP is very fast relative to other typical algorithms, at least when the measurement matrix is sufficiently large and iid Gaussian, giving some practical motivation to the effort. But, from a more practical perspective, there are serious limitations. First, if the matrix "X" is non-iid or non-zero-mean, AMP may diverge, despite the fact that the optimization objective (1.2) is convex. Second, to use AMP, we must translate the lambda parameters in (1.2) to alpha parameters somehow, and the solution proposed in (2.9) requires that the user knows the distribution of the ground truth "B". Both are unlikely to occur in practice. Now, if we accept the large iid X assumption and the AMP framework, then there are various other approaches that one could consider, and it would be good to understand how these compare to SLOPE. For example, if one knows the distribution of the ground truth, then one could use the MMSE denoiser and presumably arrive at a much better solution than SLOPE (in the sense of MSE). How much better would be interesting to know. And if the distribution of the ground truth was unknown, one could still assume a parametric distribution and tune its parameters using EM- or SURE-based AMP strategies. These two would be valid competitors of SLOPE in large iid Gaussian regime. ====================== UPDATE AFTER REBUTTAL ====================== I thought that the authors did a good job preparing the rebuttal. As they say, the main motivation of SLOPE is to perform variable selection while controlling FDR; the goal is not to minimize MSE. But my main concern, which still stands, has to do with the possibility for AMP to diverge in practice, in the case that the matrix is not drawn iid sub-Gaussian. This certainly happens in medical imaging applications and it may very well happen in some applications of interest to the statistics community. Demonstrating that it works for one application (e.g., GWAS) does not guarantee that it will work for all. One small comment is that the two non-iid-Gaussian examples in the rebuttal seem to be in the class of iid-sub-Gaussian matrices (certainly the Bernoulli example is) where AMP is already proven to work. So, those examples are not very convincing regarding universality.

[Author Response · NeurIPS 2019]

Thanks to the reviewers for the insightful and constructive feedback. It will surely improve the manuscript. Due to space constraints, instead of responding point-by-point, we address points in common with multiple reviews. All minor comments made by reviewer #1 have been addressed and incorporated into a revised version of the paper.

**i.i.d. Gaussian measurement matrix assumption.** While, in general, AMP theory provides performance guarantees only for i.i.d. sub-Gaussian data, in practice, favorable performance of AMP seems to be more universal. For example, in Fig. 1a, we illustrate the performance of AMP for i.i.d. zero mean, $1/n$ variance design matrices that are *not* Gaussian (one i.i.d. $\pm 1$ Bernoulli (top) and one i.i.d. shifted exponential (bottom)). In both cases, AMP converges very fast, thus demonstrating its robustness to distributional assumptions.

Recent work proposes a variant of AMP, called vector-AMP or VAMP, which is a computationally-efficient algorithm that *provably* works for a wide range of design matrices, namely, those that are right rotationally-invariant. We thank reviewer #2 for pointing us to "AMP for convex optimization with nonseparable penalties" by Manoel et al, which studies VAMP for a similar setting as SLOPE. However, the type of nonseparability considered in the referenced work requires the penalty to be separable on subsets of an affine transformation of its input. As such, the setting does not directly apply to SLOPE, but we have built a hybrid, "SLOPE VAMP", based on code generously shared by the authors of the referenced work, which performs very well in the (non-) i.i.d. (non-) Gaussian regime (see Fig. 1a and 1b). Motivated by these promising empirical results, we feel that theoretically understanding SLOPE dynamics with VAMP is an exciting direction that we plan to pursue in other work.

**Known signal prior assumption.** We are intrigued by reviewer #3's suggestion of using EM- or SURE-based AMP strategies to remove this assumption. We would like to pursue this within the SLOPE framework, though we haven't done so at this time. We believe that developing such strategies alongside SLOPE VAMP would provide a quite general framework for recovery of the SLOPE estimator.

**Comparison to MMSE AMP.** In general, the (statistical) motivation for using methods like LASSO or SLOPE is to perform variable selection, and in addition, for SLOPE, to control the false discovery rate. Both methods are therefore biased and, consequently, MMSE AMP strategies will, by design, outperform *if performance is based on MSE*. To combine the best of both methods, one could also incorporate a "debiasing" device in SLOPE AMP, à la "Debiasing the LASSO: optimal sample size for Gaussian designs" by Javanmard & Montanari, but we will leave this for future work. Nevertheless, Fig. 1c suggests that SLOPE AMP has MSE that is not too much worse than MMSE AMP.

**Comparison to [20] and [12].** While [20] have the same asymptotic analysis, we have a clear, rigorous statement of where it applies. That is, the analysis in [20] applies *if* the state evolution has a unique fixed point, and our Thm. 1 states precise conditions under which this is true. Moreover, we believe that our algorithmic approach offers a more concrete connection between the finite-sample behavior of the SLOPE estimator and its asymptotic distribution. We also agree with reviewer #2 that a discussion of the main results of [12] to highlight the gap between optimal estimators and any convex penalty would be useful for readers and we will add it to the final manuscript.

**Real-world data.** We agree with reviewer #1 that performing an empirical study on real-world data will significantly strengthen our results. Previous research has tested SLOPE performance on Genome-Wide Association Studies (GWAS) data (see "The Northern Finland Birth Cohort of 1966 (NFBC)"). Due to its precedence in the SLOPE literature and its inherent scientific importance, we intend to test SLOPE AMP on this data, however, there are restrictions on its use and we are currently undergoing the protocols needed to be granted access to the data by the NIH.

**(a)** i.i.d. $\pm 1$ Bernoulli design matrix (top) and i.i.d. shifted exponential design matrix (bottom)

**(b)** i.i.d. Gaussian design matrix (top) and non-i.i.d. right rotationally-invariant design matrix where AMP diverges (bottom)

**(c)** i.i.d. Gaussian design matrix

**Figure 1:** Performance of AMP variants in different settings with Bernoulli-Gaussian prior, dimension = 1000, and sample size = 300.

[Meta-Review · NeurIPS 2019]

This paper provides an AMP-based algorithm to solving SLOPE, a variant of the regularized least squares problems with a non-separable regularizer, and a proof that state evolution describes the macroscopic behavior of the proposed algorithm under the iid zero-mean Gaussian design and in the large-system limit. All the three reviewers rated this paper above the acceptance threshold. They are also satisfied with the authors’ feedback. I would thus recommend acceptance of this paper for presentation at the NeurIPS conference. Reference 20 has been published in the proceedings of the IEEE International Symposium on Information Theory, so that I would suggest updating the entry to reflect it.